

# Marine Rapid Environmental Assessment in the Gulf of Taranto: a multiscale approach

Nadia Pinardi[1,2], Vladyslav Lyubartsev[2], Nicola Cardellicchio[3], Claudio Caporale[4], Stefania Ciliberti[2], Giovanni Coppini[2], Francesca De Pascalis[6], Lorenzo Dialti[4], Ivan Federico[2], Marco Filippone[4], 5 Alessandro Grandi[5], Matteo Guideri[4], Rita Lecci[2], Lamberto Lamberti[4], Giuliano Lorenzetti[6], Paolo Lusiani[4], Cosimo Damiano Macripo'[3], Francesco Maicu[6], Diego Tartarini[4], Francesco Trotta[1], Georg Umgiesser[6], Luca Zaggia[6]

[1]Department of Physics and Astronomy, University of Bologna, Bologna, 40127, Italy
[2]Centro EuroMediterraneo sui Cambiamenti Climatici, Bologna, 40128, Italy
10 [3]Istituto per lo studio dell'Ambiente Marino Costiero-CNR, Taranto, 74100, Italy
[4]Istituto Idrografico della Marina, Genoa, 16134, Italy
[5]Istituto Nazionale di Geofisica e Vulcanologia, Bologna, 40128, Italy
[6]Istituto di Scienze Marine-CNR, Venice, 30122, Italy

15 *Correspondence to*: Nadia Pinardi (nadia.pinardi@unibo.it)

**Abstract.** A multiscale sampling experiment was carried out in the Gulf of Taranto (eastern Mediterranean) providing the first synoptic evidence of the large scale circulation structure and associated mesoscale variability. The mapping of the mesoscale and large scale geostrophic circulation showed the presence of an anticyclonic large scale Gyre occupying the central open ocean area of the Gulf of Taranto. On the periphery of the Gyre upwelling is evident where surface waters are 20 colder and saltier than at the center of the Gyre. Over a one-week period, the rim current of the Gyre undergoes large changes which are interpreted as baroclinic/barotropic instabilities, generating small scale cyclonic eddies in the periphery of the anticyclone. The eddies are generally small, one of which can be classified as a submesoscale eddy, due to its size. This eddy field modulates the upwelling regime in the Gyre periphery.

## 1 Introduction

25 Marine Rapid Environmental Assessment (MREA) was developed in the late 1990s to collect synoptic oceanographic data relevant for nowcasts, forecasts and derived applications (Robinson and Sellschopp, 2002). Situational sea awareness relies on observational and modelling information on the dynamical state of the sea in order to ensure safer and more efficient operations. MREA contributes to situational sea awareness by advancing the synoptic data acquisition for state estimation and forecasting. Opportunity observations, such as ARGO profilers, are normally not abundant enough to resolve the local 30 scales of interest especially from a synoptic point of view. Thus MREA is one of the optimal experimental strategies to collect definitive evidence on ocean mesoscales for improving knowledge and forecasts.





The ocean is eventful, intermittent and several processes generally contribute to its space-time variability. For its physical state variables, the space scales that MREA methodology targets are from a few hundred meters to several km and temporal scales range from hours to days. Assuming a several days time scale, the dominant variability is the large scale general circulation, its mesoscale and submesoscale components (Thomas et al., 2008). At shorter time scales internal waves, upper

mixed layer daily cycles and tides dominate the energy spectrum (Talley et al., 2011). MREA observations are necessary to resolve specific time and space scales of interest thus increasing the usefulness of the observations for nowcasts and forecasts.

The MREA methodology has three main components: 1) the observational strategy and ocean state estimation; 2) the

nowcast and forecasting studies; 3) the forecast/analysis validation and re-analysis. In this paper we concentrate on the observational strategy and the objective mapping of the resulting circulation structures.

MREA has been implemented globally, in several regions and adaptive sampling has also been developed (Lermusiaux, 2007, Frolov et al., 2014). Intermittency and multiscale processes have led to the concepts of nested forecasts which make

use of optimized sampling networks to increase forecast accuracy. However, it is not yet clear if a regular grid (''lawnmower'') survey, which is a "classical" strategy for synoptic ocean sampling, would be less efficient than adaptive path-planned surveys to map the synoptic variability with unknown field correlation scales.

We applied MREA to a little studied region of the world's ocean, the Gulf of Taranto in the northern Ionian Sea (Fig.1). We

use the classical, regular grid sampling strategy and explore the basic water mass properties as well as the geostrophic circulation. The insertion of these data in the large scale operational model of the Mediterranean Forecasting System (Oddo et al., 2014) and in the nested high resolution model for the Gulf of Taranto are shown in a companion paper in this issue (Federico et al., 2016).

Here we analyze four surveys carried out in the area of the Gulf of Taranto from 1 to 10 October 2014 with the Italian Navy Survey Vessel Galatea and the RV Cerruti. A new multi-scale sampling strategy was used to measure the temperature and salinity structure of the flow field from the open ocean to the shelf–coastal scales of the Gulf of Taranto (north-western Ionian Sea in the Mediterranean Sea) and the coastal-harbor scales of Mar Grande (Fig.1).

The sampling strategy and the temperature and salinity data enables us to estimate the water masses in the area and construct dynamic height and the derived geostrophic circulation using objective analysis mapping techniques. This is the first time that the circulation has been mapped for the whole Gulf of Taranto with increasing resolution from the coasts to the open ocean and with synoptic time scale resolution. The usage of the MREA14 observations to assess model performances is given in two companion papers (Federico et al., 2016, and Gaeta et al, 2016)



The paper is organized as follows: section 2 describes the data collection methodology and section 3 the water mass analysis. Section 4 presents the dynamic height and the geostrophic circulation and section 5 discusses the results.

### 2. Circulation structure and data collection methodology

The Gulf of Taranto is a deep, semi-enclosed ocean area in southern Italy encircled by two peninsulas, Salento and Calabria (Fig. 1). It is open to the northern Ionian Sea, and a deep trench of more than 2000 m connects it to the eastern Mediterranean Sea. The continental shelf area, considered as the area from the coasts to the 200 m depth contour, occupies only 10% of the total Gulf area. The shelf is wider on the Salento than the Calabria side and a 7.5 km-wide sheltered elliptical embayment, called the Mar Grande, opens in the north-eastern part of the Gulf (Fig.1).

From a large scale point of view, the mean circulation in the area can be assessed by taking the long term average of the circulation from a 20 year reanalysis (Pinardi et al., 2015). The surface circulation (Fig. 2) is generally anticyclonic in the autumn (October), while in the spring-summer, it is cyclonic (June). This opposite circulation pattern is connected to the stronger Western Adriatic Coastal Current (WACC, Guarnieri et al., 2013) entering the Gulf of Taranto from the eastern side

(Fig. 2).  One of the major aims of the MREA experiment was to verify the October circulation shown in Fig. 2.

The Gulf of Taranto is a deep semi-enclosed sea with lateral water exchanges with the Ionian Sea. The seasonally different circulation of the Gulf of Taranto described above may lead to changes in the inflow/outflow structure. In the anticyclonic case, it is likely that vertically stratified water masses enter the Gulf  from the western side (Calabria in Fig. 1) and exit from

the Salento side (eastern side, see Fig.1). We argue that Mar Grande could have both lateral and vertical exchanges, as classified by Cessi et al. (2014). The MREA experiment partly clarified these questions.

Very few CTD observations in the past have been reported in the Gulf of Taranto and none with a synoptic coverage. Our goal was mainly to carry out the first survey of the thermohaline properties of the area with synoptic coverage at three

different scales: large, shelf and harbor scale (Mar Grande).  Based on the large scale flow structure in Fig. 2, the four surveys were planned and implemented following the schemes presented in Fig. 3 and described in Table 1.

The large scale surveys (LS1 and LS2) were carried out over three days, a quasi-synoptic time scale, in an area which is on average 800 m deep. The stations were repeated in LS2 in order to understand large scale temperature and salinity changes

on a weekly basis. The mean station distance between stations was 16 km which is about the Rossby radius of deformation for the eastern Mediterranean (Hecht et al., 1988).  This spacing was chosen as a good compromise between the horizontal resolution and the time needed to cover the area synoptically.



The shelf scale survey, CS1, was carried out in the northeastern Gulf of Taranto, an extended shelf area of the Gulf (Fig.1). The mean distance between the stations was 5 km and the mean depth of the area was 400 m. The MG1 survey covers the shelf area of the Mar Grande which is a heavily human impacted harbor area. The distance between stations in the Mar
Grande is ~1 km and the mean depth was 15 m.

All measurements were carried out with Idronaut CTD 316Plus on board of the RV Galatea for LS1, CS1 and LS2 and the RV Cerruti for MG1.

### 3. Thermohaline structure of the Gulf of Taranto and Mar Grande

**3.1  Vertical structure of temperature, salinity and density**

The LS1, LS2 and CS1 mean vertical profiles are shown in Fig. 4. The temperature structure is typical of the end-of-summer stratification in the eastern Mediterranean, i.e. a mixed layer down to 30 meters and a seasonal thermocline with a temperature gradient of about 10 $^0$C. Between 100 and 300 m it is possible to detect the subsurface temperature maximum characteristic of the Modified Levantine Intermediate Water (MLIW, Theocharis et al., 1993)  which reaches slightly higher
values than 38.9 PSU in this region. The salinity at the surface does not a particular mixed layer structure, it decreases smoothly between the surface and 100 m. The low salinity values at the surface (37.8 PSU) could indicate surface waters of an Adriatic or Atlantic origin (Atlantic Modified Waters, AMW, Theocharis et al., 1993) because there are no large rivers discharging in this area. Differences between LS1, LS2 and CS1 are evident in the surface salinities: CS1 surface salinities are larger than LS1 and LS2 suggesting the upwelling of saltier waters from the subsurface. Further evidence of upwelling is
given in Section 3.2.

The interesting features of the LS1 and LS2 survey are the changes that occur over a one-week period, between the two cruises, in the first 100 m of the water column (Fig.5). LS2 was colder and fresher than LS1 by approximately 0.5 $^0$C and 0.1 PSU and the mixed layer depth had decreased by about 5 meters leading to a 0.8 $^0$C difference in temperature at 40 m (Fig.
5). These changes are likely due to a precipitation event occurring at the end of LS1, recorded in the ship logbooks.  In addition to precipitation the weather was stormy which lead to turbulent mixing in the mixed layer.

From the difference in salinity between LS1 and CS1, we can approximately compute the value of the precipitation required for such a change at the surface. Knowing that the surface water flux due to precipitation, P, amounts to a change in salt
water flux that is:



$$K_V \frac{\Delta S}{\Delta z} = -S_o P \tag{1}$$

and assuming $K_V$, the vertical diffusivity, equal to $10^{-2}\ m^2 s^{-1}$, $\Delta z = 20\ m$, $S_0 = 37.7\ PSU$, $\Delta S = -0.1$, we obtain $P = 1.32\ 10^{-6}\ ms^{-1}$ which is close to the value of $P = 2.\ 10^{-6}\ ms^{-1}$ as deduced from the atmospheric precipitation data for this period (Federico et al., 2016).

The thermocline extension is better represented by the profile of Brunt-Vaisala frequency represented in Fig. 6. Typical values are in the range of 3-15 cycles h[-1] which is relatively large for the open ocean (Talley et al., 2011) indicating that the water column is stably stratified. The maximum Brunt-Vaisala frequency is reached at a 40 m depth which is approximately the middle of the region of maximum temperature gradients in Fig. 4. Taking 3 cycles hr[-1] as the low value to mark the transition to intermediate waters, the thermocline then extends between 30 and 100 meters.

Lastly we describe the thermal and haline structure of the Mar Grande. Figure 7 shows the temperature, salinity and density structure of the water column, average from all stations. The salinity values in the first 5 meters of the water column are 0.4 PSU lower than in CS1 and LS1/LS2 indicating the source of the low salinity waters from the Mar Piccolo, located northeast of the Mar Grande (Fig.1). Between 6 and 9 meters salinity values are similar to the values in LS1/LS2, marking the entrance

of the shelf and open ocean waters from the Gulf of Taranto. As expected, the density is uniform since the harbor is a partially confined, shallow water area where turbulent mixing  renders the whole water column uniform.

### 3.2  Objective mapping of the temperature and salinity fields

In this section we describe the horizontal distribution of temperature and salinity at two different depths, one in the center of the mixed layer (10 m) and the other in the middle of the thermocline (60 m).  The mapping was carried out by an Objective

Analysis-OA technique (Bretherton, 1976, Carter and Robinson, 1987) described briefly in the Appendix.  For the mapping, CS1 and LS2 networks were combined in order to give an overall picture of the temperature and salinity structures from the shelf to the open ocean.

Fig. 8 shows the upper mixed layer temperature and salinity mapping for LS1 and LS2+CS1. During the LS1 cruise, from

October 1 to 3, 2014, the temperature and salinity mixed layer structure was dominated by a frontal structure, separating colder and saltier waters near the shelf escarpment from higher temperature, fresher waters offshore, in the open ocean areas of the Gulf of Taranto. After one week, during the CS1+LS2 cruises from October 8 to 10 2014, the temperature front



disappeared and small eddies formed. The northernmost cyclonic eddy has a diameter less than 10 km and was produced in the mapping by two stations in the CS1 network. This eddy was mixed layer intensified and it had a diameter of about one fifth of the one associated with eastern Levantine mesoscale eddies (Hecht et al., 1986).

In the seasonal thermocline, the temperature and salinity structure was dominated by a large scale anticyclonic Gyre in the center of the Gulf of Taranto (Fig.9). This Gyre was already evident in the climatological October reanalysis flow field in Fig.2. This is however the first, direct evidence of the anticyclonic structure of the Gulf of Taranto circulation for October. The anticyclone was defined by the observations as the area limited by the largest open ocean temperature and salinity gradients. It appeared that the anticyclone strengthened between LS1 and LS2, forming meanders and intensified gradient

segments. In the periphery of the anticyclonic rim current, upwelling structures are evident because the water  is colder and saltier than inside the anticyclone (Fig.9). Between LS1 and CS1+LS2 cruises the upwelled cold and salty waters on the northeaestern side of the Gulf of Taranto anticyclone had changed considerably, below and around the small scale cyclonic mixed layer eddy (Fig. 8).

In order to better represent the upwelling phenomena occurring at the periphery of the Gyre in the Gulf of Taranto, Fig. 10 shows a section of the temperature and salinity fields.  The section shows that between LS1 and LS2 the anticyclone deepened at the center and stronger upwelling occurred at its borders. The mixed layer partly re-stratified in temperature, which is consistent with submesoscale dynamics in the mixed layer (Thomas et al., 2007).

Finally the temperature and salinity mapping in the Mar Grande is shown in Fig. 11. Both surface temperature and salinity were characterized by a frontal region in the middle of the area, produced by the inflowing fresher and colder waters from the Mar Piccolo (Fig.1). Near the bottom and at mid-depth the waters were of offshore origin and upwelling was detected in the northeastern part of the area probably as part of the vertical estuarine circulation structure, as described in modeling studies (Gaeta et al., 2016).

**4. Circulation in the Gulf of Taranto in October 2014**

A Dynamic height at 10 m with respect to a reference level of 100 m was computed from the density profiles (Fig.12). This shallow reference level was chosen in order to capture part of the connections between the open ocean and the shelf.

The dynamic height shows the large scale, anticyclonic Gyre of the Gulf of Taranto in a similar position to that found in the

October climatological picture in Fig. 2. The periphery of the anticyclone was characterized by a cyclonic eddy in LS1 and by an eddy-like upwelling area in LS2. Between LS1 and LS2, the anticyclone changed noticeably in shape and intensity,



strengthening from LS1 to LS2. These changes are likely due to dynamical instabilities of the anticyclone rim current that modulate the upwelling at the periphery.

Fig. 13 shows the geostrophic velocities computed from the dynamic height. The LS1 surface currents, turning clockwise
around the anticyclonic Gyre, are characterized by jets, i.e. intensified rim current segments. In LS1 one of these jets developed between the anticyclone and the eastern side cyclonic eddy. At LS2+CS1, the Gyre underwent considerable changes in the jets strength because the rim current was meandering, which is a manifestation of baroclinic/barotropic instability and eddy growth. On the northeastern side of the anticyclonic Gyre, the mixed layer cyclonic eddy of Fig. 8 emerged as a circulation structure. The rim current of the Gyre meandered around the cylonic eddy border and a new jet
developed, with velocities of the order of 40 cm s$^{-1}$. This cyclonic eddy was much smaller than the LS1 eastern eddy, about 5-10 km in diameter, probably generated by a frontogenesis/cyclogenesys event along the anticyclonic rim current. The submesoscale field was mapped in the Mediterranean Sea (Bouffard et al., 2012) for the Balearic northern Current. Here for the first time we found the growth of a small scale, submesoscale eddy at the border of a large, sub-basin scale Gyre as a result of the meandering of its rim current. Notwithstanding the different generation mechanisms in different parts of the
basin, we believe this is an evidence of pervasive submesoscale dynamics in the Mediterranean Sea.

### 5. Discussion and conclusions

We carried out a multiscale sampling experiment in the Gulf of Taranto. Our experiment provided the first synoptic evidence of the large scale circulation structure and associated mesoscale variability. We used a classical sampling strategy, with a regularly spaced station network in three different subareas of the Gulf. In the open ocean, the LS1 and LS2 sampling was
carried out at 15 km resolution over three days while in the northeastern shelf area the station spacing was about 5 km and the 24 stations were carried out in one day. Finally the Mar Grande harbor scale was sampled at 1 km resolution and the sampling required 12 hours.

The water mass analysis for LS1, LS2 and CS1 showed that the vertical thermohaline structure in the upper 300 meters is
dominated by two water mass types: the first one belonging to the mixed layer down to 30 m characterized by relatively fresh and warm waters while the second one was recognizable as MLIW with subsurface temperature and salinity maxima. The water column is highly stratified (Brunt-Vaisala frequency of the order of 10 cycles h$^{-1}$) and the seasonal thermocline is located between 30 and 100 meters. Furthermore a precipitation event occurred between LS1 and LS2 which lowered the surface salinity of 0.1 PSU concomitantly changing the mixed layer temperatures of 0.5 $^{0}$C.




The mapping of the temperature and salinity structures was carried out by objective analysis, calibrated for the specific survey sampling scheme. Starting with the mixed layer mapping, it emerged that large scale frontal structures changed between LS1 and CS1+LS2 cruises due to the growth and decay of small scale, mixed layer intensified cyclonic eddies. The middle thermocline circulation, temperature and salinity structure of the Gulf of Taranto is dominated by an anticyclonic

5 large scale Gyre. Its periphery is dominated by upwelling as shown by the low temperature and high salinity waters, especially on the northeastern side of the Gulf of Taranto. The Gyre rim current was hydrodynamically unstable, generating frontogenesis and rim current segment intensifications. Two cyclonic eddy centers grew and decayed between LS1 and LS2. One of them could be classified as submesoscale, due to its small size, captured only by the finer sampling scheme of the CS1 survey.

The mapping of temperature and salinity fields in the Mar Grande suggests several density compensating fronts and generally low stratification, presumably connected to the estuarine vertical circulation.

In conclusion, regular sampling networks can capture most of the significant ocean variability if adequately calibrated for

15 station resolution from the open ocean to the shelf and coastal-harbor scale. More observations will be required to map the seasonal variability of the anticyclonic Gyre, the structure of the upwelling areas and their influence on the coastal ecosystem and the submesoscale flow field captured for the first time in this experiment.

20 **Acknowledgements**

This paper was partially funded by the PON Project TESSA "Technologies for Situational Sea Awareness", the Italian Ministry of Research RITMARE project. We would like to thank  Mr. Vincenzo De Palmis of IAMC-CNR for his logistic support.





**Appendix: Objective analysis mapping**

Objective Analysis is a least square estimation method also known as Gauss-Markov filter to map non regularly spaced observations into a regular grid. It was applied for the first time in oceanography by Bretherton *et al.* (1976) and Carter and Robinson (1987). The assumptions are that the statistics of the interpolated field is stationary and homogeneous.

The problem can be stated as follows: given $\varphi_r$ observations at $\vec{x}_r = (x_r, y_r)$ locations irregularly spaced with $r=1,...N$, we would like to estimate the field $\hat{\theta}_{\vec{x}}$ in a regular $\vec{x}$ grid. Assuming that $\varphi_r = \theta_r + \varepsilon_r$ where $\theta_r$ is the true field and $\varepsilon_r$ is the measurement error, the least square estimate is:

$$\hat{\theta}_{\vec{x}} = \tilde{\theta} + \sum_{r=1}^{N} C_{\vec{x}r} \left\{ \sum_{s=1}^{N} A_{rs}^{-1} \left( \varphi_s - \tilde{\theta} \right) \right\} \qquad \text{A.1}$$

The correlation functions are defined as:

$$A_{rs} = \langle \varphi_r \varphi_s \rangle = F(\vec{x}_r - \vec{x}_s) + E\delta_{rs} \qquad \text{A.2}$$

and

$$C_{\vec{x}r} = \langle \theta_{\vec{x}} \varphi_r \rangle = F(\vec{x} - \vec{x}_r) \qquad \text{A.3}$$

where brackets indicate the ensemble mean. *F* is commonly written as:

$$F(r) = \left( 1 - \frac{r^2}{b^2} \right) e^{-\frac{r^2}{2a^2}} \qquad \text{A.4}$$

with $r^2 = x^2 + y^2$ being the square of the distance between two points and $a,b$ the decorrelation and decay lengths respectively. $E$ is the measurement error variance, $\langle \varepsilon_r \varepsilon_s \rangle = E\delta_{rs}$ which is taken in all our calculation to be 10% of the field variance, assuming to be dominated by the representativeness error.





$\tilde{\theta}$ is the field mean estimated from the observations using the weighted average:

$$\tilde{\theta} = \frac{1}{\sum_{r,s}^{N} A_{rs}^{-1}} \sum_{r,s}^{N} A_{rs}^{-1} \varphi_s \qquad \text{A.5}$$

The correlation functions used for mapping depends on the observational sampling, i.e. sparse observations will need to consider relatively larger decorrelation scales with respect to denser networks which will require smaller values of $a$. The $a,b$ values used in this paper are listed in Table A1 and different ones were chosen for the three sampling schemes used, for LS1, LS2, CS1 and MG1. A radius of influence is also used within which the $N$ observations are chosen for each estimated grid point which is also listed in Table A1.

The interpolated field percentage error variance is written as (Bretherton, 1976):

$$\frac{\left(\theta_{\bar{x}} - \hat{\theta}_{\bar{x}}\right)^2}{C_{xx}} = 1 - \frac{\sum_{r,s}^{N} C_{xr} A_{rs}^{-1} C_{xs}}{C_{xx}} + \frac{\left(1 - \sum_{r,s}^{N} C_{xs} A_{sr}\right)^2}{C_{xx} \sum_{r,s}^{N} A_{rs}^{-1}} \qquad \text{A.6}$$

The square root of A.6 is given in Fig. A.1 for the LS1, LS2+CS1 and MG1 surveys. The growth of errors from the station points outward depends on the correlation parameters $a,b$ in A.4. All our mapped fields were masked in order not to display areas with errors larger than 50% for LS1, LS2+CS1 and larger than 20% for MG1 .




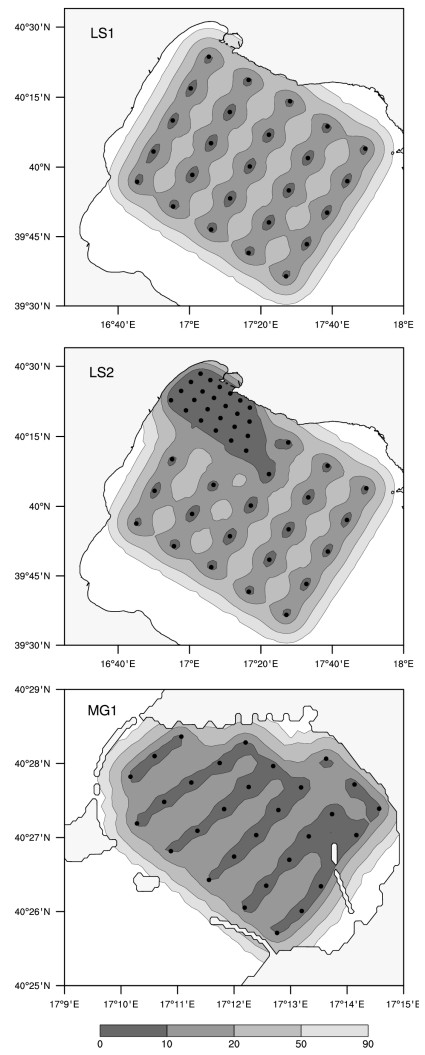

**Fig. A1.** Objective analysis percentage error field for the three surveys using the parameters of Table A1 (units are %).





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





| Cruise name | Number of CTD stations | Data collection time period (days) | Start, end dates and time |
|---|---|---|---|
| LS1 | 26 | 3 | Oct. 1 at 15:30 to Oct. 3 at 24:00 |
| MG1 | 31 | 1 | October 5 from 7:15 to 11:30 |
| CS1 | 24 | 1 | Oct 8 from 16:00 to 20:20 |
| LS2 | 25 | 3 | Oct. 9 from 01:00 to Oct. 10 at 24:00 |

5 **Table 1:** number of CTD stations carried out during the surveys at large, shelf and harbor scales.

| Survey | Grid size (degrees) | a (km) | b (km) | Radius of influence |
|---|---|---|---|---|
| MG1 | 1/512 (~200 m) | 2 | 1 | 1 km |
| LS1 & LS2+CS1 | 1/32 (~3 km) | 25 | 15 | 20 km |

10 **Table A1:** Objective analysis mapping parameters




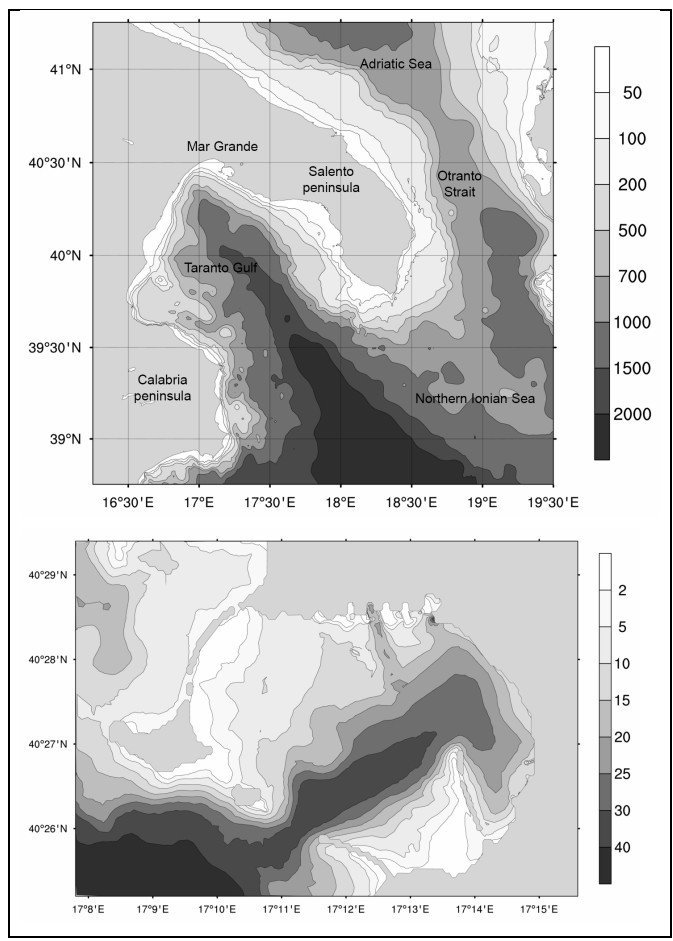

**Fig. 1** The Gulf of Taranto (upper panel) and the Mar Grande (lower panel) bathymetry and coastlines.





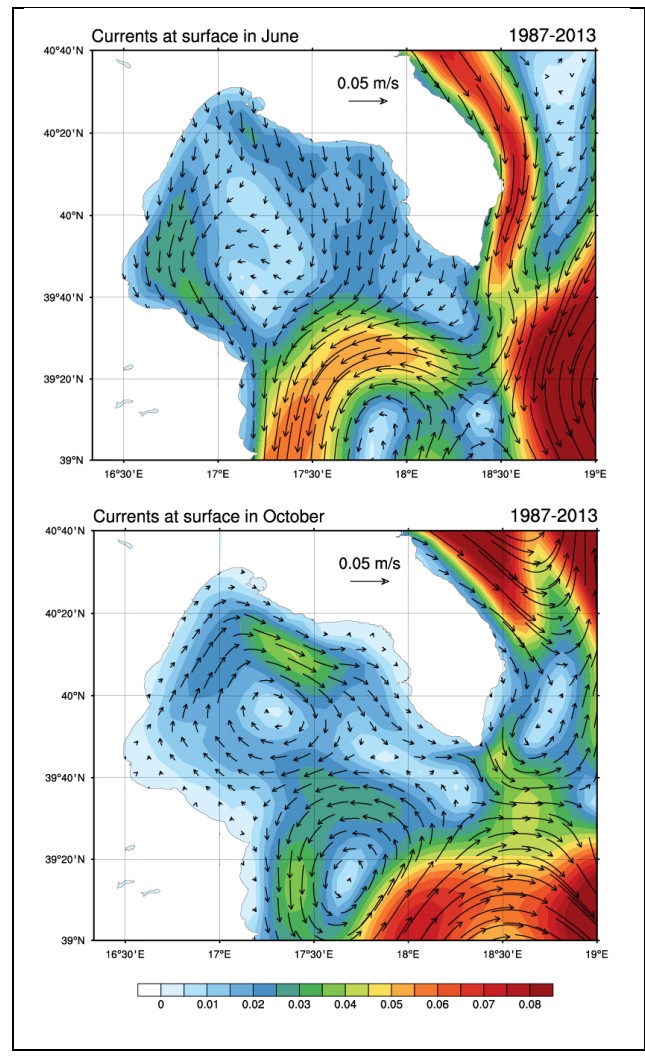

**Fig. 2** Monthly mean 1987-2013 surface currents from reanalysis (Pinardi et al., 2015) in the Gulf of Taranto. Top panel: June. Bottom panel: October. The units are m s$^{-1}$ and the color indicates amplitude





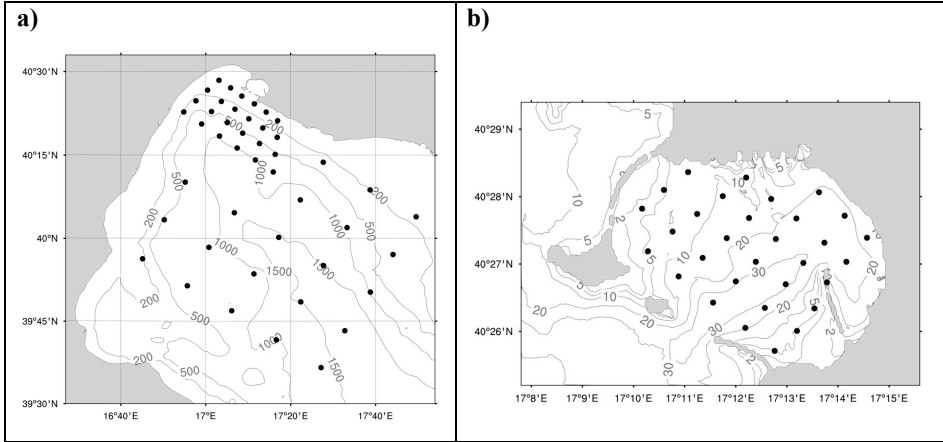

5   **Fig. 3** The four survey CTD station distributions: a) the large scale survey LS1 and LS2 with the coastal scale (CS1) survey
in the northwestern corner; b) the Mar Grande (MG) survey.





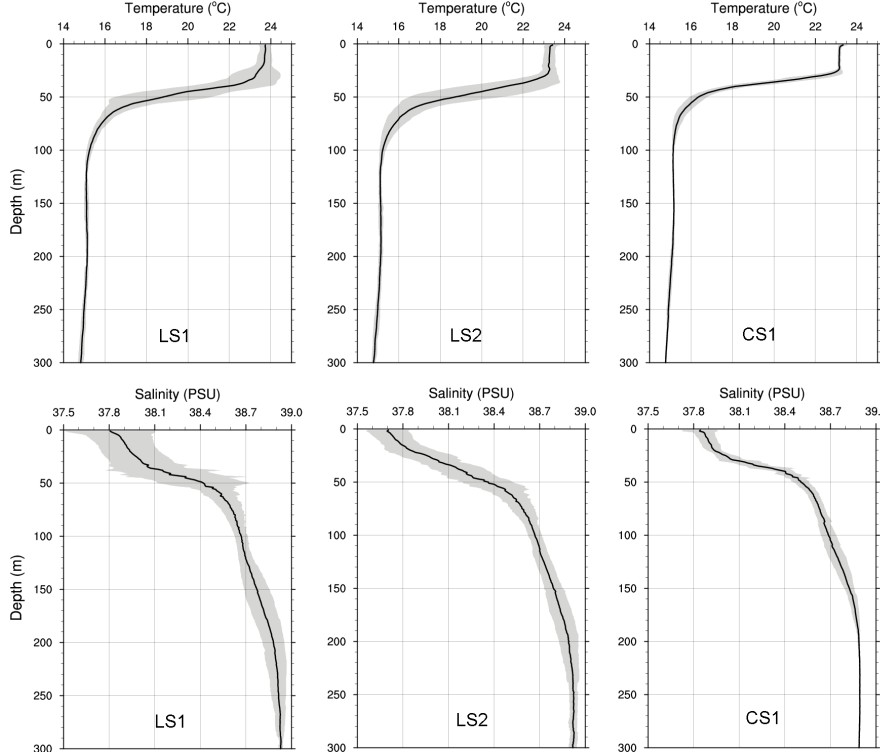

**Fig. 4** The station average temperature and salinity profiles for LS1 (left column), LS2 (central) and CS1 (right column) surveys. The grey shaded areas represent standard deviations of the survey station profiles from the mean.





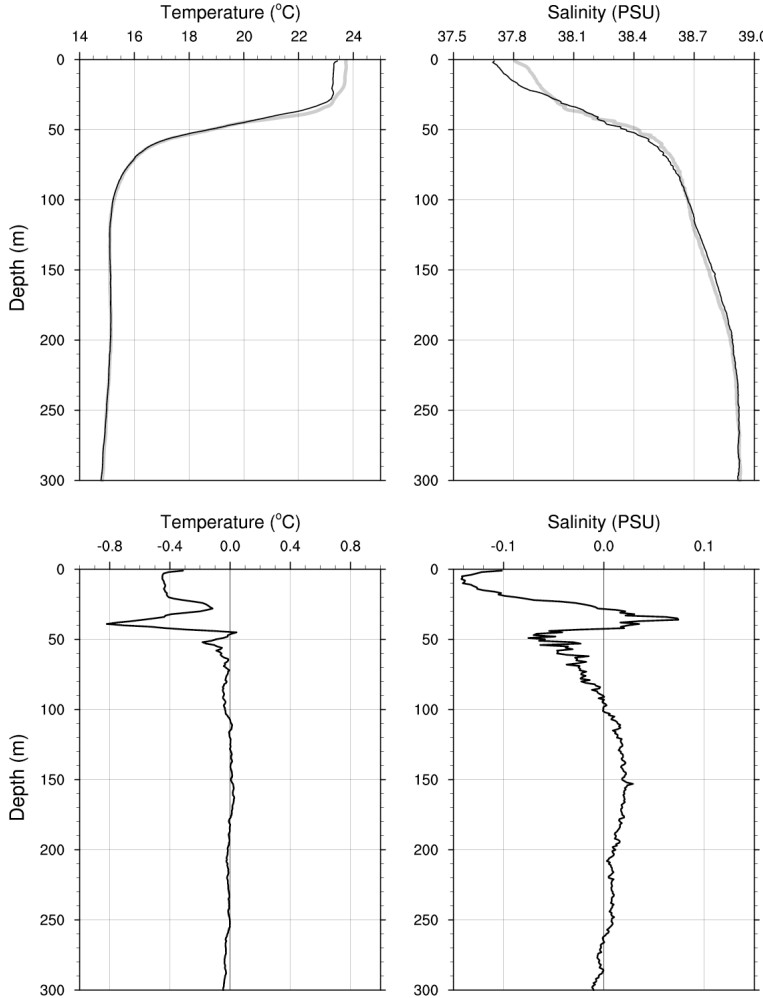

**Fig. 5** Superimposed  mean temperature and salinity LS1 and LS2 profiles and differences. Top left: LS1 (grey line) and LS2

5   (black line) mean temperature profiles. Top right: LS1 (grey line) and LS2 (black line) mean salinity profiles. Bottom left:

    LS2-LS1 mean temperature difference. Bottom right: LS2-LS1 mean salinity difference.





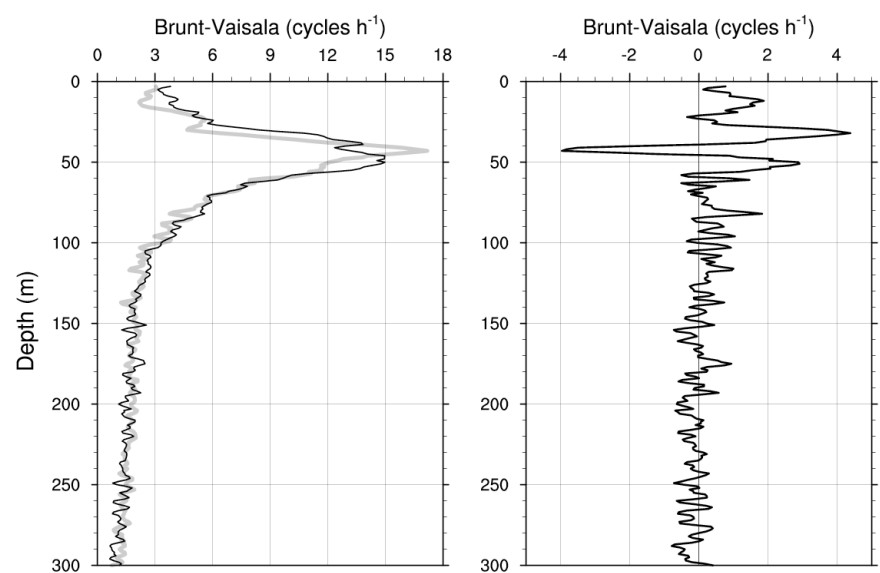

**Fig. 6** Left Panel: Mean Brunt-Vaisala frequency (cycles hr$^{-1}$) for LS1 (grey) and LS2 (black) surveys. Right Panel: LS2 minus LS1 Brunt-Vaisala profile.





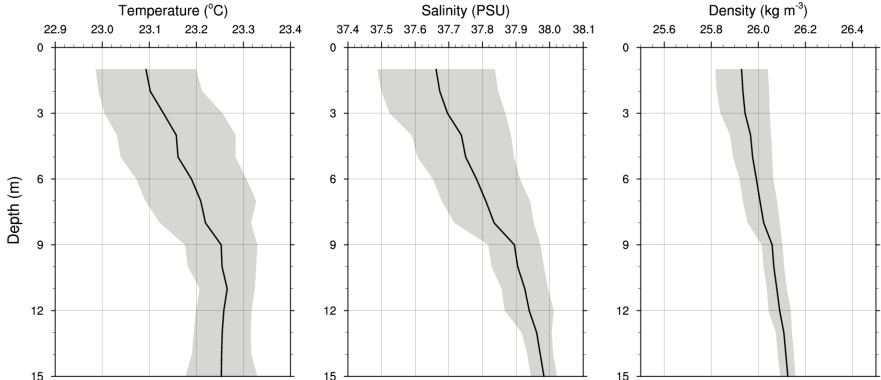

**Fig. 7** Mar Grande horizontal average temperature (left), salinity (center) and density (right) profiles for October 5, 2014.

5   The grey shaded areas represent standard deviations from the mean.




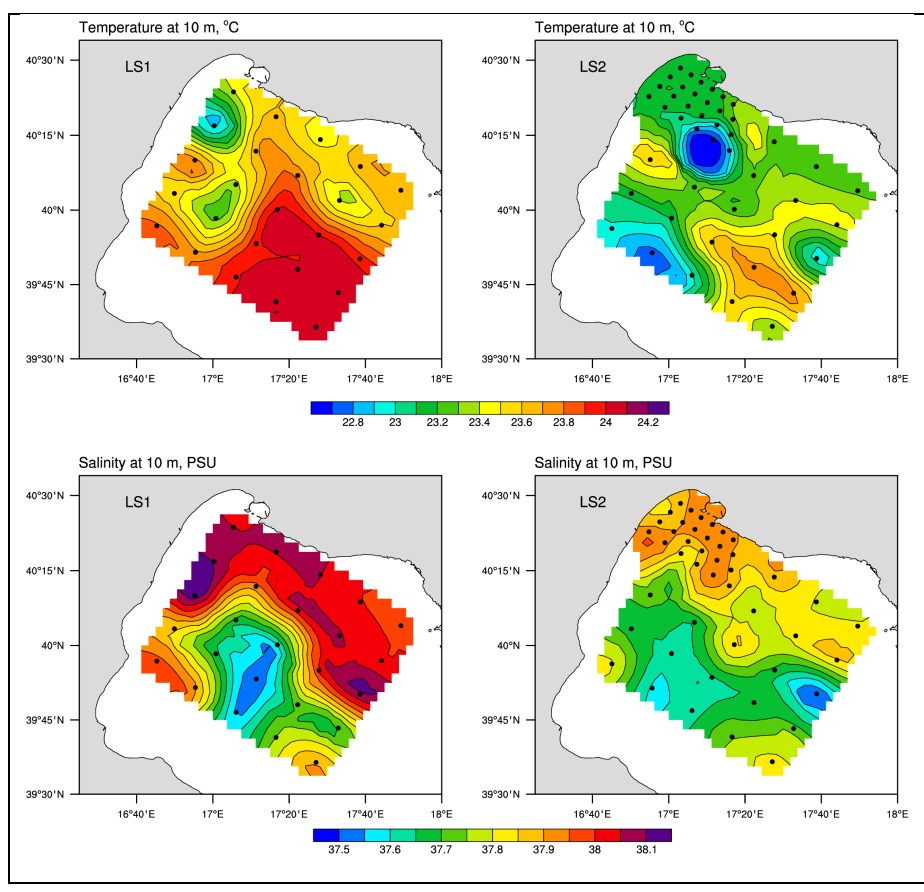

**Fig. 8** Temperature and salinity objective mapping at 10 m. Left top and bottom panel: LS1 temperature and salinity fields. Right top and bottom panel: LS2 temperature and salinity fields.





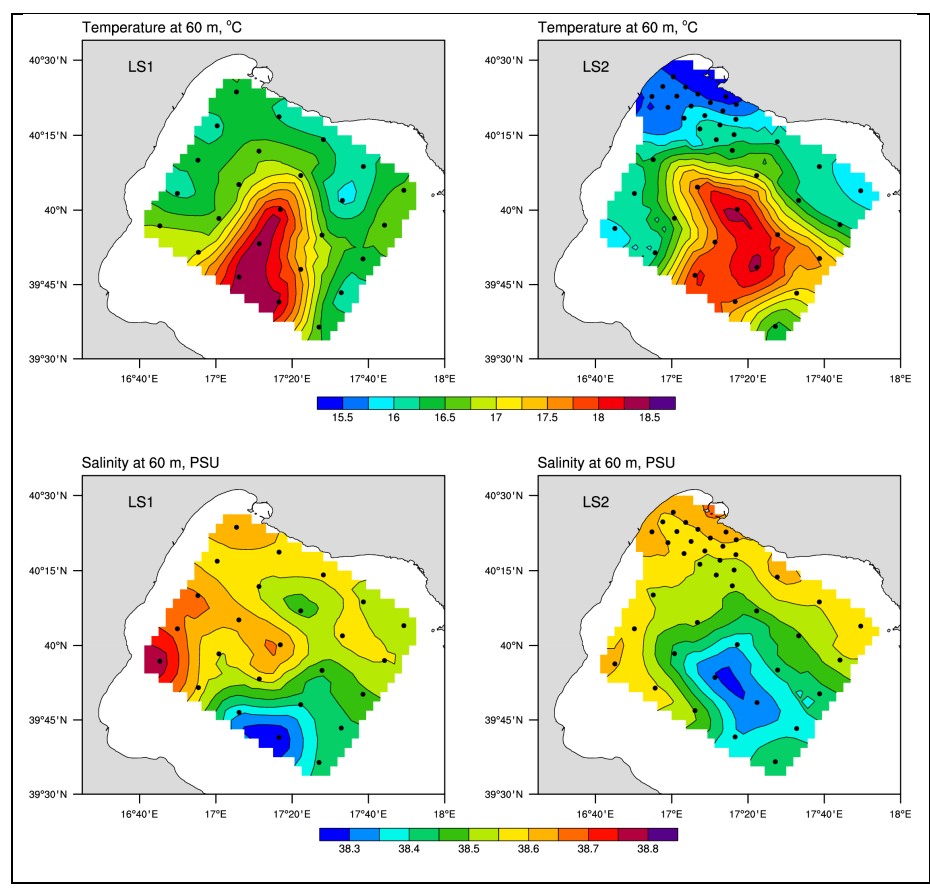

5 **Fig. 9** Seasonal thermocline (60 m) temperature (top) and salinity (bottom) mapping for LS1 (left panel) and LS2 (right panel) .





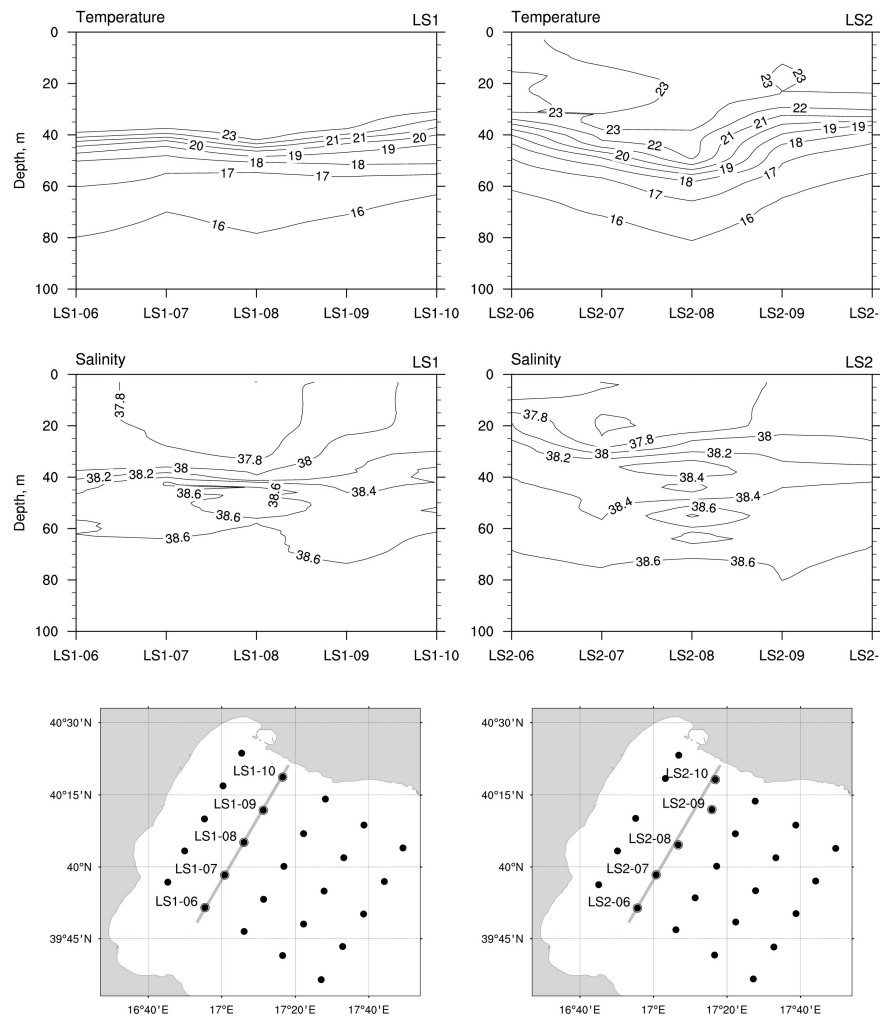

5 **Fig. 10** Temperature (top) and salinity (intermediate) transects for LS1 (left) and LS2 (right). The LS1 and LS2 transect stations are illustrated in the two bottom panels.





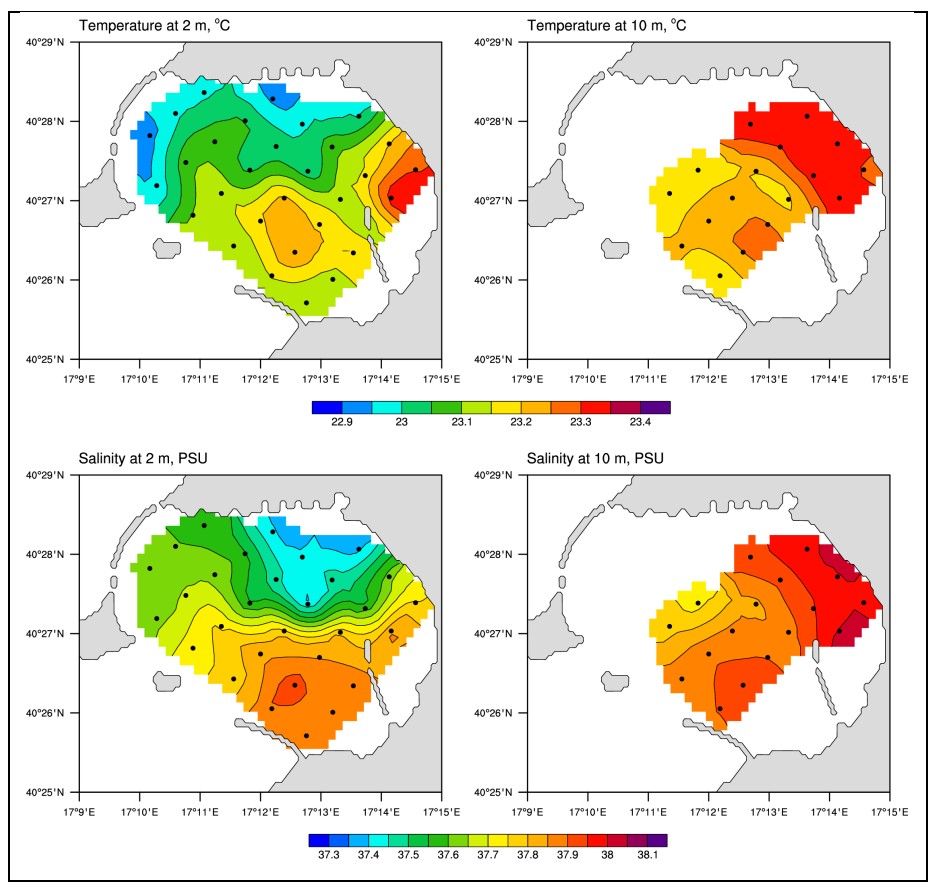

**Fig. 11** Mar Grande near surface (2 m) and near bottom (10 m) mapping of temperature and salinity fields for October 5,
5   2014.




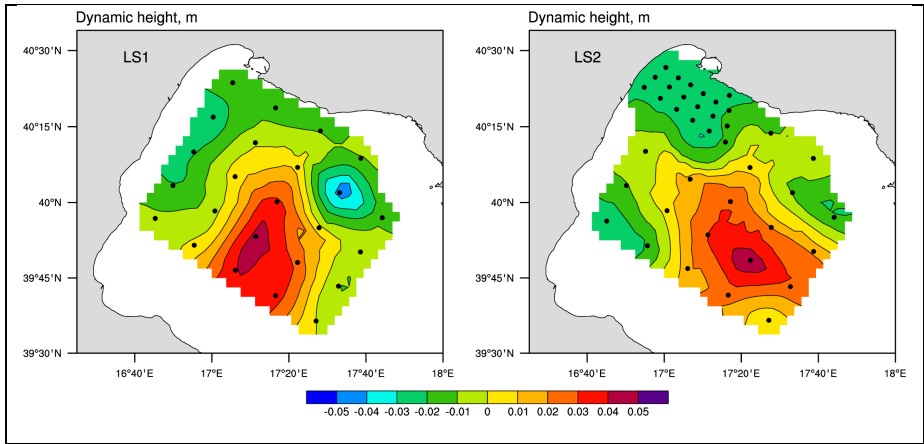

**Fig. 12** Dynamic height at 10 m with respect to 100 m reference level. Left panel: LS1. Right panel: LS2+CS1

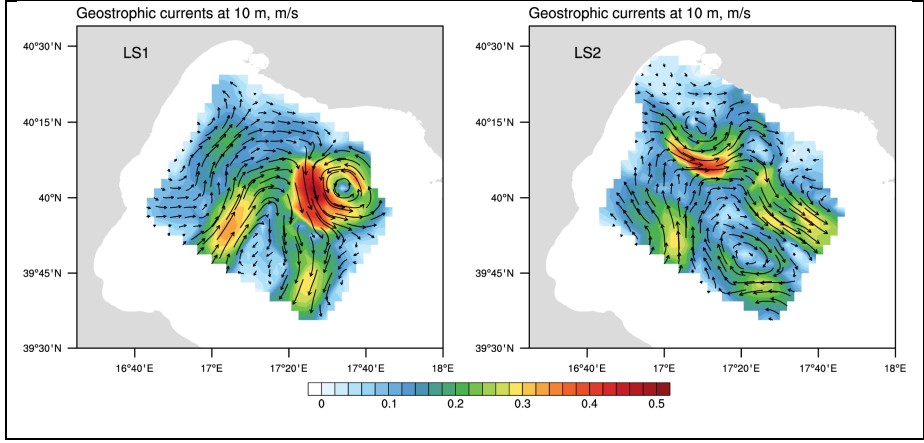

**Fig. 13** Geostrophic current velocities at 10 m referred to 100 m for LS1 (left top panel), LS2 (right top panel).

