# Peer review of "Marine Rapid Environmental Assessment in the Gulf of Taranto: a multiscale approach"

_Natural Hazards and Earth System Sciences, 2016_

## Referee Comment (RC1) · Anonymous Referee #1 · 8 Jul 2016

The manuscript describes a oceanographic sampling experiment carried out to in the Gulf of Taranto in the Northern Ionian Sea. Four oceanographic campaigns were performed during around 10 days in October 2015, measuring T and Salinity profiles in both open ocean and coastal areas for a set of sampling stations covering the whole area with different spatial resolution. This multi scale approach has been followed to describe by means of a synoptic set of measurements both the large scale and the meso-scale circulation in the Gulf. The obtained results highlight the effectiveness of this approach as a possible standard procedure to be followed in case of MREA for this area.

The paper falls within the scope of the journal. The arguments treated are very interesting and promising, nevertheless a moderate revision of the manuscript is required before being published.

[Figure]

As a general comment, I suggest to do not only concentrate on the observational strategy and the obtained results but also to deepen the discussion on the usefulness of the specific aforementioned strategy in view of both a support to operational oceanography and to the management of the emergencies at sea (pollution, S&R etc.,). Some comments should be specifically included into the Discussion section.

Following, some specific comments and suggestions are provided:

Page 3, line 20: "We argue that .. The MREA experiment partially clarified these questions". In the following, any paragraphs specifically dealing with this issue are not found. Some information should be added in the results sections or, if not fundamental with respect to the scope of the paper, maybe the sentence should be changed.

Page 3 and figure 3: I suggest to use different colors to highlight the sampling points of each survey.

Page 4: how do you compute the average ? The adopted procedure should be explicated.

Page 4, line 13: "the subsurface temperature maximum" probably it is the salinity.

Page 4: some more details about the meteorological conditions during the 10 days measurements, e.g the wind intensity and directions, could help with the comprehension of the results. Especially when comparing the LS1 and LS2 sampling results.

Page 5: paragraph from line 16 to line 21 need to be rephrased, it is not completely clear.

Page 5, line 26: why only CS1 and LS2 were combined together ? It is better to explain.

Page 5 and figure 8: I suggest to include in the panels some number or letters to identify properly the dynamic structures.

Page 6: in the left top panel, in the northwestern corner of the image, the small gyre subsequently observed in right top panel seems already existing or at least starting.

Also the dynamic eigth analysis and the geostrophic velocity fields (figure 13) seem to put into evidence. I am not sure this could be a reasonable evidence or just a speculation.

---

## Referee Comment (RC2) · Anonymous Referee #2 · 28 Jul 2016

MS No.: nhess-2016-179 MS Type: Research article Special Issue: Situational sea awareness technologies for maritime safety and marine environment protection

Title: Marine Rapid Environmental Assessment in the Gulf of Taranto: a multiscale approach By N. Pinardi, V. Lyubartsev, N. Cardellicchio, C. Caporale, S. Ciliberti, G. Coppini, F. De Pascalis, L. Dialti, I. Federico, M. Filippone, A. Grandi, M. Guideri, R. Lecci, L. Lamberti, G. Lorenzetti, P. Lusiani, C. D. Macripo', F. Maicu, D. Tartarini, F. Trotta, G. Umgiesser, and L. Zaggia

The manuscript "Marine Rapid Environmental Assessment in the Gulf of Taranto: a multiscale approach" by Pinardi et al., is about a multiscale sampling experiment carried out in the Gulf of Taranto to collect synoptic oceanographic data over a 10-days period and subsequently study the thermohaline structure and the geostrophic circula-

tion of the area and its variability. The data analysis from the four surveys carried out in the area from 1 to 10 October 2014 provides evidence of the large scale circulation structure and associated mesoscale variability of the area consisting of an anticyclonic large scale Gyre that occupies the central open ocean area of the Gulf of Taranto and a rim current on the periphery of the gyre that undergoes large changes over the 10-days period. Overall, I found the manuscript and the work very interesting and I think that it deserves publication to the NHESS journal after some minor revisions are made according to the following comments:

General comments:

The design strategy, the methodology and the usefulness of the MREA experiment conducted in the Gulf of Taranto should be discussed more with emphasis on the usability of the MREA experiment results. In the present version of the manuscript, the authors concentrate only on the analysis of the observational data collected during the oceanographic surveys. I am also expecting to see an introductory discussion about the scientific purpose (i.e. which are the scientific questions to be answered) of conducting a multiscale MREA experiment in the Gulf of Taranto. At the same time, the authors should try to enrich the discussion session of the manuscript by adding more discussion on their scientific findings and the perspectives of their work.

Specific comments:

1. Introduction section: "A new multi-scale sampling strategy was used ...... the coastal-harbor scales of Mar Grande (Fig.1).

The authors should explain better the novelty of the approach adopted to measure the T/S structure of the Gulf of Taranto.

2. Circulation structure and data collection methodology: "From a large scale point of view . . . in Fig. 2"

As the Ionian basin circulation has undergone significant interannual changes over
the period 1987 – 2013 the authors should present more evidence that the long term average of this period is representative of the hydrodynamic situation in the Gulf of Taranto in June and October.

3. T/S diagrams could be useful to depict the water mass structure of the area.

4. A more detailed discussion on the instability of the rim current is expected.

5. "Furthermore a precipitation event occurred between LS1 and LS2 which lowered the surface salinity of 0.1 PSU concomitantly changing the mixed layer temperatures of 0.5 C"

The authors should explain better how the precipitation event changed the mixed layer temperatures by 0.5 C

---

## Author Comment (AC3) · 1 Nov 2016

[revised manuscript text omitted]

The paper is organized as follows: section 2 describes the data collection methodology and section 3 the water mass analysis. Section 4 presents the dynamic height and the geostrophic circulation and section 5 discusses the results.

**2. Circulation structure and data collection methodology**

The Gulf of Taranto is a deep, semi-enclosed ocean area in southern Italy encircled by two peninsulas, Salento and Calabria (Fig. 1). It is open to the northern Ionian Sea, and a deep trench of more than 2000 m connects it to the eastern Mediterranean Sea. The continental shelf area, considered as the area from the coasts to the 200 m depth contour, occupies only 10% of the total Gulf area. The shelf is wider on the Salento than the Calabria side and a 7.5 km-wide sheltered elliptical embayment, called the Mar Grande, opens in the north-eastern part of the Gulf (Fig.1).

From a large scale point of view, the mean circulation in the area can be assessed by taking the current fields from a reanalysis product (Pinardi et al., 2015) that does not contain the MREA data. The surface circulation (Fig. 2) is anticyclonic in October 2014, while in June 2014 it is cyclonic. This opposite circulation pattern is probably connected to the different Western Adriatic Coastal Current (WACC, Guarnieri et al., 2013), Northern Ionian Sea outflow/inflow system in the two months and the local atmospheric forcing. One of the major aims of the MREA experiment was to verify the October circulation shown in Fig. 2.

The Gulf of Taranto is a deep semi-enclosed sea with lateral water exchanges with the Ionian Sea. The seasonally different circulation of the Gulf of Taranto described above may lead to changes in the inflow/outflow structure. In the anticyclonic case, it is likely that vertically stratified water masses enter the Gulf from the western side (Calabria in Fig. 1) and exit from the Salento side (eastern side, see Fig.1). We argue that Mar Grande could have both lateral and vertical exchanges, as classified by Cessi et al. (2014). The MREA experiment partly clarified these questions.

Very few CTD observations in the past have been reported in the Gulf of Taranto and none with a synoptic coverage. Our goal was mainly to carry out the first survey of the thermohaline properties of the area with synoptic coverage at three

nadia pinardi 10/29/2016 7:57 PM

nadia pinardi 10/29/2016 7:57 PM

nadia pinardi 10/29/2016 7:57 PM

nadia pinardi 10/29/2016 7:57 PM

nadia pinardi 10/29/2016 7:58 PM

nadia pinardi 10/30/2016 6:46 AM

nadia pinardi 10/29/2016 7:58 PM

nadia pinardi 10/29/2016 7:59 PM

nadia pinardi 10/29/2016 7:59 PM

nadia pinardi 10/30/2016 6:46 AM

nadia pinardi 10/30/2016 6:47 AM

different scales: large, shelf and harbor scale (Mar Grande).  Based on the large scale flow structure in Fig. 2, the four surveys were planned and implemented following the schemes presented in Fig. 3 and described in Table 1.

The large scale surveys (LS1 and LS2) were carried out over three days, a quasi-synoptic time scale, in an area which is on average 800 m deep. The stations were repeated in LS2 in order to understand large scale temperature and salinity changes on a weekly basis. The mean station distance between stations was 16 km which is about the Rossby radius of deformation for the eastern Mediterranean (Hecht et al., 1988).  This spacing was chosen as a good compromise between the horizontal resolution and the time needed to cover the area synoptically.

The shelf scale survey, CS1, was carried out in the northeastern Gulf of Taranto, an extended shelf area of the Gulf (Fig.1). The mean distance between the stations was 5 km and the mean depth of the area was 400 m. The MG1 survey covers the shelf area of the Mar Grande which is a heavily human impacted harbor area. The distance between stations in the Mar Grande is ~1 km and the mean depth was 15 m.

All measurements were carried out with Idronaut CTD 316Plus on board of the RV Galatea for LS1, CS1 and LS2 and the RV Cerruti for MG1.

**3. Thermohaline structure of the Gulf of Taranto and Mar Grande**

**3.1  Vertical structure of temperature, salinity and density**

The LS1, LS2 and CS1 mean vertical profiles are shown in Fig. 4. The mean profile is estimated by taking the arithmetic average of observational points across the profiles which are defined on a 1 m regular vertical grid. The temperature structure is typical of the end-of-summer stratification in the eastern Mediterranean, i.e. a mixed layer down to 30 meters and a seasonal thermocline with a temperature gradient of about 10 $^0$C. Between 100 and 300 m it is possible to detect the subsurface salinity maximum characteristic of the Modified Levantine Intermediate Water (MLIW, Theocharis et al., 1993) which reaches slightly higher values than 38.9 PSU in this region. The salinity at the surface does not have a particular mixed layer structure, it decreases smoothly between the surface and 100 m. The low salinity values at the surface (37.8 PSU) could indicate surface waters of an Adriatic or Atlantic origin (Atlantic Modified Waters, AMW, Theocharis et al., 1993) because there are no large rivers discharging in this area. Differences between LS1, LS2 and CS1 are evident in the surface salinities: CS1 surface salinities are larger than LS1 and LS2 suggesting the upwelling of saltier waters from the subsurface. Further evidence of upwelling is given in Section 3.2.

Fig. 5 shows a T-S diagram of the LS1, LS2 profiles to better identify the water masses and types. Some of the profiles extended to 900 m depth, in the central Gulf of Taranto trench (Fig.1) so that four water masses can be detected, one more

nadia pinardi 10/29/2016 6:55 PM

with respect the three already discussed for the first 300 m. The first water mass is the surface water mass, indicated by water type 1 in Fig. 5, corresponding to low salinity and almost constant temperature. The second is the thermocline water type (number 2 in Fig. 5), due to the mixing of the surface waters and MLIW as shown clearly by the clustering of the T-S points around a line joining the two water types. Furthermore, MLIW (point 3 in Fig. 5) is now clearly detectable with a salinity and temperature increase with respect to the thermocline water mass type. Finally a deep water mass type (4 in figure 5) is also evident, with temperatures lower than 14 C and relatively low salinities, probably of Adriatic origin.

The interesting features of the LS1 and LS2 survey are the changes that occur over a one-week period, between the two cruises, in the first 100 m of the water column (Fig. 6). LS2 was colder and fresher than LS1 by approximately 0.5 $^0$C and 0.1 PSU and the mixed layer depth had decreased by about 5 meters leading to a 0.8 $^0$C difference in temperature at 40 m (Fig. 6). The weather conditions deteriorated after October 4 and large winds developed on October 5 while precipitation started October 3 and continued up to October 5 (Fig. 7). Such atmospheric forcing changes can justify the temperature and salinity decrease at the surface, as discussed below.

[revised manuscript text omitted]

nadia pinardi 10/30/2016 8:30 AM

nadia pinardi 10/30/2016 8:30 AM

nadia pinardi 10/30/2016 8:30 AM

nadia pinardi 10/30/2016 8:30 AM

nadia pinardi 10/30/2016 8:30 AM

nadia pinardi 10/30/2016 8:32 AM

The instability of gyre rim currents and/or large mesoscale eddy field borders has been studied in the past (Mc Williams et al., 1983, Pinardi et al., 1987, Staneva et al., 2001) and more recently for submesoscale generating fronts (Hamlington et al., 2014). The instabilities of rim currents connected to temperature frontal structures generate eddies, which are due to cyclogenenetic processes such as mixed baroclinic/barotropic instabilities. In our case the observations show that instabilities occur in a week long time and most importantly modulate the upwelling phenomena at the open ocean-shelf areas interface, a mechanism that could be very important to support good environmental conditions in the near coastal regions. Numerical modelling studies have now started to understand the vorticity and energy dynamics of the flow field observed in this experiment.

In conclusion, regular sampling networks can capture most of the significant ocean variability if adequately calibrated for station resolution from the open ocean to the shelf and coastal-harbor scale. More observations will be required to map the seasonal variability of the anticyclonic Gyre, the structure of the upwelling areas and their influence on the coastal ecosystem and the submesoscale flow field captured for the first time in this experiment.

This paper MREA sampling methodology could be also used to collect data in order to respond to environmental emergencies, such as oil spills or other pollutant dispersal. If the location of the source of pollution is known, the CS1 sampling strategy could be carried out in one day, and forecasting models adjusted to the measured fields through data assimilation, improving the forecast skill. Thus this paper has also put the basis for a protocol of in situ data collection that could support emergency management at sea.

**Acknowledgements**

This paper was partially funded by the PON Project TESSA "Technologies for Situational Sea Awareness", the Italian Ministry of Research RITMARE project. We would like to thank  Mr. Vincenzo De Palmis of IAMC-CNR for his logistic support.

**Appendix: Objective analysis mapping**

Objective Analysis is a least square estimation method also known as Gauss-Markov filter to map non regularly spaced observations into a regular grid. It was applied for the first time in oceanography by Bretherton *et al.* (1976) and Carter and Robinson (1987). The assumptions are that the statistics of the interpolated field is stationary and homogeneous.

The problem can be stated as follows: given $\varphi_r$ observations at $\vec{x}_r = (x_r, y_r)$ locations irregularly spaced with $r=1,...N$, we would like to estimate the field $\hat{\theta}_{\vec{x}}$ in a regular $\vec{x}$ grid. Assuming that $\varphi_r = \theta_r + \varepsilon_r$ where $\theta_r$ is the true field and $\varepsilon_r$ is the measurement error, the least square estimate is:

[revised manuscript text omitted]

McWilliams, J.C., E.D. Brown, H.L. Bryden, C.C. Ebbesmeyer, B.A. Elliot, R.H. Heinmiller, B.L.Hua, K.D. Leaman, E.J. Lindstrom, J.R. Luyten, S.E. McDowell, W.B. Owens, H. Perkins, J.F.Price, L. Regier, S.C. Riser, H.T. Rossby, T.B.
10  Sanford, C.Y. Shen, B.A. Taft, & J.C. Van Leer, 1983. The local dynamics of eddies in the western North Atlantic. In: Eddies in Marine Science, A.R. Robinson, ed., Springer-Verlag, Berlin Heidelberg, 92-113.

Oddo, P., A. Bonaduce, N. Pinardi, and A. Guarnieri, 2014. Sensitivity of the Mediterranean sea level to atmospheric pressure and free surface elevation numerical formulation in NEMO. Geosci. Model Dev., 7, 3001–3015. doi:10.5194/gmd-
15  7-3001-2014

N. Pinardi and A. R. Robinson, 1987. Dynamics of deep thermocline jets in the Polymode Region. Journal of Physical Oceanography, Vol. 17(8), pp.1163-1188, doi:10.1175/1520-0485(1987)017<1163:DODTJI>2.0.CO;2

20  Pinardi, N., M. Zavatarelli, M. Adani, G. Coppini, C. Fratianni, P. Oddo, S. Simoncelli, M.Tonani, V. Lyubartsev, S. Dobricic, 2015. The Mediterranean Sea large scale low frequency ocean variability from 1987 to 2007: a retrospective analysis, Progress in Oceanography, doi: 10.1016/j.pocean.2013.11.003

Robinson, A.R. and J. Shellschopp, 2002. Rapid Assessment of the Coastal Ocean Environment. In: Pinardi N. and Woods J.
25  (Eds), Ocean Forecasting, Springer-verlag.

Staneva, J.V., D.E. Dietrich, E. V. Stanev, M. J. Bowman, 2001. Rim current and coastal eddy mechanisms in an eddy-resolving Black Sea general circulation model. Journal of Marine Systems 31 2001 137–157.

30  Talley, L., D., G.L. Pickard, W.J. Emery, J.H. Swift, 2011. Descriptive Physical Oceanography: an introduction. Academic Press, Elsevier.

Theocharis, A., D. Georgopoulos, A. Lascaratos, K. Nittis, 1993. Water masses and circulation in the central region of the Eastern Mediterranean: Eastern Ionian, South Aegean and Northwest Levantine, 1986–1987. Deep Sea Research-Part II, 40 (6), 1121-1142, doi:10.1016/0967-0645(93)90064-T

5   Thomas, L. N., Tandon, A. and Mahadevan, A., 2008. Submesoscale Processes and Dynamics, in Ocean Modeling in an Eddying Regime (eds M. W. Hecht and H. Hasumi), American Geophysical Union, Washington, D. C.. doi: 10.1029/177GM04

| Cruise name | Number of CTD stations | Data collection time period (days) | Start, end dates and time |
|---|---|---|---|
| LS1 | 26 | 3 | Oct. 1 at 15:30 to Oct. 3 at 24:00 |
| MG1 | 31 | 1 | October 5 from 7:15 to 11:30 |
| CS1 | 24 | 1 | Oct 8 from 16:00 to 20:20 |
| LS2 | 25 | 3 | Oct. 9 from 01:00 to Oct. 10 at 24:00 |

5 **Table 1:** number of CTD stations carried out during the surveys at large, shelf and harbor scales.

| Survey | Grid size (degrees) | a (km) | b (km) | Radius of influence |
|---|---|---|---|---|
| MG1 | 1/512 (~200 m) | 2 | 1 | 1 km |
| LS1 & LS2+CS1 | 1/32 (~3 km) | 25 | 15 | 20 km |

10 **Table A1:** Objective analysis mapping parameters

[Figure]

**Fig. 1** The Gulf of Taranto (upper panel) and the Mar Grande (lower panel) bathymetry and coastlines. In the lower panel, the connection of Mar Grande with Mar Piccolo is indicated.

[Figure]

**Fig. 2** Monthly mean surface currents from reanalysis (Pinardi et al., 2015) in the Gulf of Taranto. Top panel: June 2014.
Bottom panel: October 2014. The units are m s$^{-1}$ and the color indicates the amplitude

[Figure]

**Fig. 3** The four survey CTD station distributions: a) the large scale survey LS1 and LS2 (red bullets) with the coastal scale (CS1) survey in the northwestern corner (blue bullets); b) the Mar Grande (MG) survey.

[Figure]

**Fig. 4** The station average temperature and salinity profiles for LS1 (left column), LS2 (central) and CS1 (right column) surveys. The grey shaded areas represent standard deviations of the survey station profiles from the mean.

[Figure]

**Fig. 5** T-S diagram for all the LS1-LS2 profiles, covering the depths of 1-900 m (the deepest point is in the central trench of the Gulf of Taranto, see Fig. 1). Numbers refer to the 4 water mass types found in the profiles and discussed in the text.

nadia pinardi 10/30/2016 7:01 AM

nadia pinardi 10/30/2016 7:00 AM

[Figure]

**Fig.** 6 Superimposed mean temperature and salinity LS1 and LS2 profiles and differences. Top left: LS1 (grey line) and LS2 (black line) mean temperature profiles. Top right: LS1 (grey line) and LS2 (black line) mean salinity profiles. Bottom left: LS2-LS1 mean temperature difference. Bottom right: LS2-LS1 mean salinity difference.

nadia pinardi 10/30/2016 7:18 AM

[Figure]

**Fig. 7** Area average precipitation and 10 m wind magnitude during the cruise period from a limited area high resolution weather forecasting model of the Mediterranean Sea (Bonavita et al., 2008). Precipitation is visualized as an histogram (units of m s⁻¹) and wind magnitude is the red curve (units of m s⁻¹).

[Figure]

**Fig. 8** Left Panel: Mean Brunt-Vaisala frequency (cycles hr$^{-1}$) for LS1 (grey) and LS2 (black) surveys. Right Panel: LS2 minus LS1 Brunt-Vaisala profile.

nadia pinardi 10/29/2016 7:04 PM

[Figure]

**Fig. 9.** Mar Grande horizontal average temperature (left), salinity (center) and density (right) profiles for October 5, 2014. The grey shaded areas represent standard deviations from the mean.

nadia pinardi 10/29/2016 7:04 PM

[Figure]

5 **Fig. 10** Temperature and salinity objective mapping at 10 m. Left top and bottom panel: LS1 temperature and salinity fields. Right top and bottom panel: LS2 temperature and salinity fields. Symbols indicate four cold core eddies (C1, C2, C3, C4), the temperature and salinity front (F1) and the upwelling area (UP).

[Figure]

5    **Fig. 11.** Seasonal thermocline (60 m) temperature (top) and salinity (bottom) mapping for LS1 (left panel) and LS2 (right panel) .

nadia pinardi 10/29/2016 7:04 PM

[Figure]

5   **Fig. 12**, Temperature (top) and salinity (intermediate) transects for LS1 (left) and LS2 (right). The LS1 and LS2 transect stations are illustrated in the two bottom panels.

nadia pinardi 10/29/2016 7:04 PM

[Figure]

**Fig. 13** Mar Grande near surface (2 m) and near bottom (10 m) mapping of temperature and salinity fields for October 5, 2014.

nadia pinardi 10/29/2016 7:04 PM

[Figure]

**Fig. 14** Dynamic height at 10 m with respect to 100 m reference level. Left panel: LS1. Right panel: LS2+CS1

nadia pinardi 10/29/2016 7:04 PM

[Figure]

**Fig. 15** Geostrophic current velocities at 10 m referred to 100 m for LS1 (left top panel), LS2 (right top panel).

nadia pinardi 10/29/2016 7:04 PM

---

## Author Comment (AC4) · 1 Nov 2016

[revised manuscript text omitted]

The paper is organized as follows: section 2 describes the data collection methodology and section 3 the water mass analysis. Section 4 presents the dynamic height and the geostrophic circulation and section 5 discusses the results.

**2. Circulation structure and data collection methodology**

The Gulf of Taranto is a deep, semi-enclosed ocean area in southern Italy encircled by two peninsulas, Salento and Calabria (Fig. 1). It is open to the northern Ionian Sea, and a deep trench of more than 2000 m connects it to the eastern Mediterranean Sea. The continental shelf area, considered as the area from the coasts to the 200 m depth contour, occupies only 10% of the total Gulf area. The shelf is wider on the Salento than the Calabria side and a 7.5 km-wide sheltered elliptical embayment, called the Mar Grande, opens in the north-eastern part of the Gulf (Fig.1).

From a large scale point of view, the mean circulation in the area can be assessed by taking the current fields from a reanalysis product (Pinardi et al., 2015) that does not contain the MREA data. The surface circulation (Fig. 2) is anticyclonic in October 2014, while in June 2014 it is cyclonic. This opposite circulation pattern is probably connected to the different Western Adriatic Coastal Current (WACC, Guarnieri et al., 2013), Northern Ionian Sea outflow/inflow system in the two months and the local atmospheric forcing.. One of the major aims of the MREA experiment was to verify the October circulation shown in Fig. 2.

The Gulf of Taranto is a deep semi-enclosed sea with lateral water exchanges with the Ionian Sea. The seasonally different circulation of the Gulf of Taranto described above may lead to changes in the inflow/outflow structure. In the anticyclonic case, it is likely that vertically stratified water masses enter the Gulf from the western side (Calabria in Fig. 1) and exit from the Salento side (eastern side, see Fig.1). We argue that Mar Grande could have both lateral and vertical exchanges, as classified by Cessi et al. (2014). The MREA experiment partly clarified these questions.

Very few CTD observations in the past have been reported in the Gulf of Taranto and none with a synoptic coverage. Our goal was mainly to carry out the first survey of the thermohaline properties of the area with synoptic coverage at three

different scales: large, shelf and harbor scale (Mar Grande). Based on the large scale flow structure in Fig. 2, the four surveys were planned and implemented following the schemes presented in Fig. 3 and described in Table 1.

The large scale surveys (LS1 and LS2) were carried out over three days, a quasi-synoptic time scale, in an area which is on average 800 m deep. The stations were repeated in LS2 in order to understand large scale temperature and salinity changes on a weekly basis. The mean station distance between stations was 16 km which is about the Rossby radius of deformation for the eastern Mediterranean (Hecht et al., 1988). This spacing was chosen as a good compromise between the horizontal resolution and the time needed to cover the area synoptically.

The shelf scale survey, CS1, was carried out in the northeastern Gulf of Taranto, an extended shelf area of the Gulf (Fig.1). The mean distance between the stations was 5 km and the mean depth of the area was 400 m. The MG1 survey covers the shelf area of the Mar Grande which is a heavily human impacted harbor area. The distance between stations in the Mar Grande is ~1 km and the mean depth was 15 m.

All measurements were carried out with Idronaut CTD 316Plus on board of the RV Galatea for LS1, CS1 and LS2 and the RV Cerruti for MG1.

**3. Thermohaline structure of the Gulf of Taranto and Mar Grande**

**3.1 Vertical structure of temperature, salinity and density**

The LS1, LS2 and CS1 mean vertical profiles are shown in Fig. 4. The mean profile is estimated by taking the arithmetic average of observational points across the profiles which are defined on a 1 m regular vertical grid. The temperature structure is typical of the end-of-summer stratification in the eastern Mediterranean, i.e. a mixed layer down to 30 meters and a seasonal thermocline with a temperature gradient of about 10 $^0$C. Between 100 and 300 m it is possible to detect the subsurface salinity maximum characteristic of the Modified Levantine Intermediate Water (MLIW, Theocharis et al., 1993) which reaches slightly higher values than 38.9 PSU in this region. The salinity at the surface does not have a particular mixed layer structure, it decreases smoothly between the surface and 100 m. The low salinity values at the surface (37.8 PSU) could indicate surface waters of an Adriatic or Atlantic origin (Atlantic Modified Waters, AMW, Theocharis et al., 1993) because there are no large rivers discharging in this area. Differences between LS1, LS2 and CS1 are evident in the surface salinities: CS1 surface salinities are larger than LS1 and LS2 suggesting the upwelling of saltier waters from the subsurface. Further evidence of upwelling is given in Section 3.2.

Fig. 5 shows a T-S diagram of the LS1, LS2 profiles to better identify the water masses and types. Some of the profiles extended to 900 m depth, in the central Gulf of Taranto trench (Fig.1) so that four water masses can be detected, one more

with respect the three already discussed for the first 300 m. The first water mass is the surface water mass, indicated by water type 1 in Fig. 5, corresponding to low salinity and almost constant temperature. The second is the thermocline water type (number 2 in Fig. 5), due to the mixing of the surface waters and MLIW as shown clearly by the clustering of the T-S points around a line joining the two water types. Furthermore, MLIW (point 3 in Fig. 5) is now clearly detectable with a salinity and temperature increase with respect to the thermocline water mass type. Finally a deep water mass type (4 in figure 5) is also evident, with temperatures lower than 14 C and relatively low salinities, probably of Adriatic origin.

The interesting features of the LS1 and LS2 survey are the changes that occur over a one-week period, between the two cruises, in the first 100 m of the water column (Fig.6). LS2 was colder and fresher than LS1 by approximately 0.5 $^0$C and 0.1 PSU and the mixed layer depth had decreased by about 5 meters leading to a 0.8 $^0$C difference in temperature at 40 m (Fig. 6). The weather conditions deteriorated after October 4 and large winds developed on October 5 while precipitation started October 3 and continued up to October 5 (Fig. 7). Such atmospheric forcing changes can justify the temperature and salinity decrease at the surface, as discussed below.

From the difference in salinity between LS1 and CS1, we can approximately compute the value of the precipitation required for such a change at the surface. Knowing that the surface water flux due to precipitation, P, amounts to a change in salt water flux that is:

$$K_V \frac{\Delta S}{\Delta z} = -S_o P \qquad (1)$$

and assuming $K_V$, the vertical diffusivity, equal to $10^{-3} \, m^2 s^{-1}$, $\Delta z = 20 \, m$, $S_0 = 37.7 \, PSU$, $\Delta S = -0.1$, we obtain $P = 1.3 \, 10^{-7} \, ms^{-1}$ which is close to the time average value of precipitation, $P = 1.4 \, 10^{-7} \, ms^{-1}$ shown in Fig. 7 for this period..

The thermocline extension is better represented by the profile of Brunt-Vaisala frequency represented in Fig. 8. Typical values are in the range of 3-15 cycles h[-1] which is relatively large for the open ocean (Talley et al., 2011) indicating that the water column is stably stratified. The maximum Brunt-Vaisala frequency is reached at a 40 m, which is approximately the middle depth of the region of maximum temperature gradients in Fig. 4 and a depth located within the thermocline water mass layer shown in Fig. 5. Taking 3 cycles hr[-1] as the low value to mark the transition to intermediate waters, the thermocline then extends between 30 and 100 meters.

Lastly we describe the thermal and haline structure of the Mar Grande. Figure 9 shows the temperature, salinity and density structure of the water column, average from all stations. The salinity values in the first 5 meters of the water column are 0.4 PSU lower than in CS1 and LS1/LS2 indicating the source of the low salinity waters from the Mar Piccolo, located northeast of the Mar Grande (Fig.1). Between 6 and 9 meters salinity values are similar to the values in LS1/LS2, marking the entrance

5    of the shelf and open ocean waters from the Gulf of Taranto. As expected, the density is uniform since the harbor is a partially confined, shallow water area where turbulent mixing renders the whole water column uniform.

**3.2 Objective mapping of the temperature and salinity fields**

In this section we describe the horizontal distribution of temperature and salinity at two different depths, one in the center of the mixed layer (10 m) and the other in the middle of the thermocline (60 m). The mapping was carried out by an Objective

10   Analysis-OA technique (Bretherton, 1976, Carter and Robinson, 1987) described briefly in the Appendix. For the mapping, CS1 and LS2 networks were combined in order to give an overall picture of the temperature and salinity structures from the shelf to the open ocean. CS1 was merged only with LS2 since it was taken at the beginning of the LS2 survey and the combination was still synoptic.

15   Fig. 10 shows the upper mixed layer temperature and salinity mapping for LS1 and LS2+CS1. During the LS1 cruise, from October 1 to 3, 2014, the temperature and salinity mixed layer structure was dominated by a frontal structure (F1), separating colder and saltier waters near the shelf escarpment from higher temperature, fresher waters offshore, in the open ocean areas of the Gulf of Taranto. Three cold core eddies (C1, C2 and C3 in Fig. 10) are present north of the frontal structure. After one week, during the CS1+LS2 cruises from October 8 to 10 2014, the temperature front disappeared and smaller eddies formed.

[revised manuscript text omitted]

The instability of gyre rim currents and/or large mesoscale eddy field borders has been studied in the past (Mc Williams et al., 1983, Pinardi et al., 1987, Staneva et al., 2001) and more recently for submesoscale generating fronts (Hamlington et al., 2014). The instabilities of rim currents connected to temperature frontal structures generate eddies, which are due to cyclogenenetic processes such as mixed baroclinic/barotropic instabilities. In our case the observations show that instabilities occur in a week long time and most importantly modulate the upwelling phenomena at the open ocean-shelf areas interface, a mechanism that could be very important to support good environmental conditions in the near coastal regions. Numerical modelling studies have now started to understand the vorticity and energy dynamics of the flow field observed in this experiment.

In conclusion, regular sampling networks can capture most of the significant ocean variability if adequately calibrated for station resolution from the open ocean to the shelf and coastal-harbor scale. More observations will be required to map the seasonal variability of the anticyclonic Gyre, the structure of the upwelling areas and their influence on the coastal ecosystem and the submesoscale flow field captured for the first time in this experiment.

This paper MREA sampling methodology could be also used to collect data in order to respond to environmental emergencies, such as oil spills or other pollutant dispersal. If the location of the source of pollution is known, the CS1 sampling strategy could be carried out in one day, and forecasting models adjusted to the measured fields through data assimilation, improving the forecast skill. Thus this paper has also put the basis for a protocol of in situ data collection that could support emergency management at sea.

**Acknowledgements**

This paper was partially funded by the PON Project TESSA "Technologies for Situational Sea Awareness", the Italian Ministry of Research RITMARE project. We would like to thank Mr. Vincenzo De Palmis of IAMC-CNR for his logistic support.

**Appendix: Objective analysis mapping**

Objective Analysis is a least square estimation method also known as Gauss-Markov filter to map non regularly spaced observations into a regular grid. It was applied for the first time in oceanography by Bretherton *et al.* (1976) and Carter and Robinson (1987). The assumptions are that the statistics of the interpolated field is stationary and homogeneous.

The problem can be stated as follows: given $\varphi_r$ observations at $\vec{x}_r = (x_r, y_r)$ locations irregularly spaced with *r=1,...N*, we would like to estimate the field $\hat{\theta}_{\vec{x}}$ in a regular $\vec{x}$ grid. Assuming that $\varphi_r = \theta_r + \varepsilon_r$ where $\theta_r$ is the true field and $\varepsilon_r$ is the measurement error, the least square estimate is:

[revised manuscript text omitted]

McWilliams, J.C., E.D. Brown, H.L. Bryden, C.C. Ebbesmeyer, B.A. Elliot, R.H. Heinmiller, B.L.Hua, K.D. Leaman, E.J. Lindstrom, J.R. Luyten, S.E. McDowell, W.B. Owens, H. Perkins, J.F.Price, L. Regier, S.C. Riser, H.T. Rossby, T.B.
10   Sanford, C.Y. Shen, B.A. Taft, & J.C. Van Leer, 1983. The local dynamics of eddies in the western North Atlantic. In: Eddies in Marine Science, A.R. Robinson, ed., Springer-Verlag, Berlin Heidelberg, 92-113.

Oddo, P., A. Bonaduce, N. Pinardi, and A. Guarnieri, 2014. Sensitivity of the Mediterranean sea level to atmospheric pressure and free surface elevation numerical formulation in NEMO. Geosci. Model Dev., 7, 3001–3015. doi:10.5194/gmd-
15   7-3001-2014

N. Pinardi and A. R. Robinson, 1987. Dynamics of deep thermocline jets in the Polymode Region. Journal of Physical Oceanography, Vol. 17(8), pp.1163-1188, doi:10.1175/1520-0485(1987)017<1163:DODTJI>2.0.CO;2

20   Pinardi, N., M. Zavatarelli, M. Adani, G. Coppini, C. Fratianni, P. Oddo, S. Simoncelli, M.Tonani, V. Lyubartsev, S. Dobricic, 2015. The Mediterranean Sea large scale low frequency ocean variability from 1987 to 2007: a retrospective analysis, Progress in Oceanography, doi: 10.1016/j.pocean.2013.11.003

Robinson, A.R. and J. Shellschopp, 2002. Rapid Assessment of the Coastal Ocean Environment. In: Pinardi N. and Woods J.
25   (Eds), Ocean Forecasting, Springer-verlag.

Staneva, J.V., D.E. Dietrich, E. V. Stanev, M. J. Bowman, 2001. Rim current and coastal eddy mechanisms in an eddy-resolving Black Sea general circulation model. Journal of Marine Systems 31 2001 137–157.

30   Talley, L., D., G.L. Pickard, W.J. Emery, J.H. Swift, 2011. Descriptive Physical Oceanography: an introduction. Academic Press, Elsevier.

Theocharis, A., D. Georgopoulos, A. Lascaratos, K. Nittis, 1993. Water masses and circulation in the central region of the Eastern Mediterranean: Eastern Ionian, South Aegean and Northwest Levantine, 1986–1987. Deep Sea Research-Part II, 40 (6), 1121-1142, doi:10.1016/0967-0645(93)90064-T

5  Thomas, L. N., Tandon, A. and Mahadevan, A., 2008. Submesoscale Processes and Dynamics, in Ocean Modeling in an Eddying Regime (eds M. W. Hecht and H. Hasumi), American Geophysical Union, Washington, D. C.. doi: 10.1029/177GM04

| Cruise name | Number of CTD stations | Data collection time period (days) | Start, end dates and time |
|---|---|---|---|
| LS1 | 26 | 3 | Oct. 1 at 15:30 to Oct. 3 at 24:00 |
| MG1 | 31 | 1 | October 5 from 7:15 to 11:30 |
| CS1 | 24 | 1 | Oct 8 from 16:00 to 20:20 |
| LS2 | 25 | 3 | Oct. 9 from 01:00 to Oct. 10 at 24:00 |

5 **Table 1:** number of CTD stations carried out during the surveys at large, shelf and harbor scales.

| Survey | Grid size (degrees) | a (km) | b (km) | Radius of influence |
|---|---|---|---|---|
| MG1 | 1/512 (~200 m) | 2 | 1 | 1 km |
| LS1 & LS2+CS1 | 1/32 (~3 km) | 25 | 15 | 20 km |

10 **Table A1:** Objective analysis mapping parameters

[Figure]

5 **Fig. 1** The Gulf of Taranto (upper panel) and the Mar Grande (lower panel) bathymetry and coastlines. In the lower panel, the connection of Mar Grande with Mar Piccolo is indicated.

[Figure]

**Fig. 2** Monthly mean surface currents from reanalysis (Pinardi et al., 2015) in the Gulf of Taranto. Top panel: June 2014. Bottom panel: October 2014. The units are m s[-1] and the color indicates the amplitude

[Figure]

**Fig. 3** The four survey CTD station distributions: a) the large scale survey LS1 and LS2 (red bullets) with the coastal scale (CS1) survey in the northwestern corner (blue bullets); b) the Mar Grande (MG) survey.

[Figure]

**Fig. 4** The station average temperature and salinity profiles for LS1 (left column), LS2 (central) and CS1 (right column) surveys. The grey shaded areas represent standard deviations of the survey station profiles from the mean.

[Figure]

**Fig. 5** T-S diagram for all the LS1-LS2 profiles, covering the depths of 1-900 m (the deepest point is in the central trench of the Gulf of Taranto, see Fig. 1). Numbers refer to the 4 water mass types found in the profiles and discussed in the text.

[Figure]

**Fig. 6** Superimposed  mean temperature and salinity LS1 and LS2 profiles and differences. Top left: LS1 (grey line) and LS2 (black line) mean temperature profiles. Top right: LS1 (grey line) and LS2 (black line) mean salinity profiles. Bottom left: LS2-LS1 mean temperature difference. Bottom right: LS2-LS1 mean salinity difference.

[Figure]

**Fig. 7** Area average precipitation and 10 m wind magnitude during the cruise period from a limited area high resolution weather forecasting model of the Mediterranean Sea (Bonavita et al., 2008). Precipitation is visualized as an histogram (units of m s[-1]) and wind magnitude is the red curve (units of m s[-1]).

[Figure]

**Fig. 8** Left Panel: Mean Brunt-Vaisala frequency (cycles hr[-1]) for LS1 (grey) and LS2 (black) surveys. Right Panel: LS2 minus LS1 Brunt-Vaisala profile.

[Figure]

**Fig. 9** Mar Grande horizontal average temperature (left), salinity (center) and density (right) profiles for October 5, 2014. The grey shaded areas represent standard deviations from the mean.

[Figure]

**Fig. 10** Temperature and salinity objective mapping at 10 m. Left top and bottom panel: LS1 temperature and salinity fields. Right top and bottom panel: LS2 temperature and salinity fields. Symbols indicate four cold core eddies (C1, C2, C3, C4), the temperature and salinity front (F1) and the upwelling area (UP).

[Figure]

5 **Fig. 11** Seasonal thermocline (60 m) temperature (top) and salinity (bottom) mapping for LS1 (left panel) and LS2 (right panel) .

[Figure]

5    **Fig. 12** Temperature (top) and salinity (intermediate) transects for LS1 (left) and LS2 (right). The LS1 and LS2 transect

stations are illustrated in the two bottom panels.

[Figure]

**Fig. 13** Mar Grande near surface (2 m) and near bottom (10 m) mapping of temperature and salinity fields for October 5, 2014.

[Figure]

**Fig. 14** Dynamic height at 10 m with respect to 100 m reference level. Left panel: LS1. Right panel: LS2+CS1

[Figure]

**Fig. 15** Geostrophic current velocities at 10 m referred to 100 m for LS1 (left top panel), LS2 (right top panel).

---

## Author Comment (AC1)

Review of nhess-2016-179

Title: Marine Rapid Environmental Assessment in the Gulf of Taranto: a multiscale approach

by Pinardi et al.

In the following we have listed the referee comments and immediately after our response, with an explicit reference to the insertion of the revised texts and figures.

**Referee 1**
The manuscript describes a oceanographic sampling experiment carried out to in the Gulf of Taranto in the Northern Ionian Sea. Four oceanographic campaigns were performed during around 10 days in October 2015, measuring T and Salinity profiles in both open ocean and coastal areas for a set of sampling stations covering the whole area with different spatial resolution. This multi scale approach has been followed to describe by means of a synoptic set of measurements both the large scale and the meso-scale circulation in the Gulf. The obtained results highlight the effectiveness of this approach as a possible standard procedure to be followed in case of MREA for this area.
The paper falls within the scope of the journal. The arguments treated are very interesting and promising, nevertheless a moderate revision of the manuscript is required before being published.
**Authors**
We thank the referee for his/her appreciation of our work.

**Referee**
As a general comment, I suggest to do not only concentrate on the observational strategy and the obtained results but also to deepen the discussion on the usefulness of the specific aforementioned strategy in view of both a support to operational oceanography and to the management of the emergencies at sea (pollution, S&R etc.,). Some comments should be specifically included into the Discussion section.
**Authors**
We have added a sentence in the discussion section that reads as follows:
This paper MREA sampling methodology could be also used to collect data in order to respond to environmental emergencies, such as oil spills or other pollutant dispersal. If the location of the source of pollution is known, the CS1 sampling strategy could be carried out in one day, and forecasting models adjusted to the measured fields through data assimilation, improving the forecast skill. Thus this paper has also put the basis for a protocol of in situ data collection that could support emergency management at sea.

**Referee**
Page 3, line 20: "We argue that .. The MREA experiment partially clarified these questions". In the following, any paragraphs specifically dealing with this issue are not found. Some information should be added in the results sections or, if not fundamental with respect to the scope of the paper, maybe the sentence should be changed.
**Authors**
The referee is correct, we left the question unanswered. We added a sentence in section 3 where we discuss the objective mapping of salinity and temperature fields in the Mar Grande. After line 10 of page 7 we then added the following sentence:
The MREA strategy in the Mar Grande finally elucidated the estuarine nature of the circulation in the Mar Grande at unprecedented resolution.

**Referee**
Page 3 and figure 3: I suggest to use different colors to highlight the sampling points
of each survey.
**Authors**
We have changed Fig. 3 and used a color code, see new picture below.

[Figure]

**Referee**
Page 4: how do you compute the average? The adopted procedure should be explicated.
**Authors**
The post-processed observation profiles are given in 1 m regular vertical grid so the average
is the arithmetic mean at each vertical point. We have added after line 19 of page 4 the
following sentence:
The mean profile is estimated by taking the arithmetic average of observational points across
the profiles, which are defined on a 1 m regular vertical grid.

**Referee**
Page 4, line 13: "the subsurface temperature maximum" probably it is the salinity.
**Authors**
The referee is correct we substituted temperature with salinity, thanks.

**Referee**
Page 4: some more details about the meteorological conditions during the 10 days
measurements, e.g. the wind intensity and directions, could help with the comprehension
of the results. Especially when comparing the LS1 and LS2 sampling results.
**Authors**
We have added a new figure 7 on the wind magnitude and precipitation conditions for the 10
days of the surveys. The new Fig. 7 is reproduced below and a sentence has been added at the
new page 5 that reads as follows:
" The weather conditions deteriorated after October 4 and large winds developed on October
5 while precipitation started October 3 and continued up to October 5. Such atmospheric
forcing changes can justify the temperature and salinity decrease at the surface, as discussed
below."

[Figure]

**Fig. 7** Area average precipitation and 10 m wind magnitude during the cruise period from a limited area high resolution weather forecasting model of the Mediterranean Sea (Bonavita et al., 2008). Precipitation is visualized as an histogram (units of m s$^{-1}$) and wind magnitude is the red curve (units of m s$^{-1}$).

**Referee**
Page 5: paragraph from line 16 to line 21 need to be rephrased, it is not completely clear.
**Authors**
We realized that in Fig.1 there is no indication of the Mar Piccolo channels so we modified Fig. 1. We believe this was the problem.

**Referee**
Page 5, line 26: why only CS1 and LS2 were combined together? It is better to explain.
**Authors**
CS1 was taken at the beginning of LS2 so in order to maintain a synoptic data collection we merge it with LS2. We explain this now after line 27, page 6 now, by saying:
CS1 was merged only with LS2 since it was taken at the beginning of the LS2 survey and the combination was still synoptic.

**Referee**
Page 5 and figure 8: I suggest to include in the panels some number or letters to identify properly the dynamic structures.
**Authors**
We re-plotted Fig. 8, now Fig. 10, indicating the frontal structure and the eddies with letters, making reference to them in the text. The new Fig 10 is reproduced here. Text has been modified at page 6 to refer to symbols in Fig. 10.

[Figure]

**Fig. 10** Temperature and salinity objective mapping at 10 m. Left top and bottom panel: LS1 temperature and salinity fields. Right top and bottom panel: LS2 temperature and salinity fields. Symbols indicate four cold core eddies (C1, C2, C3, C4), the temperature and salinity front (F1) and the upwelling area (UP).

**Referee**

Page 6: in the left top panel, in the northwestern corner of the image, the small gyre subsequently observed in right top panel seems already existing or at least starting. Also the dynamic height analysis and the geostrophic velocity fields (figure 13) seem
to put into evidence. I am not sure this could be a reasonable evidence or just a speculation.

**Authors**

We do not believe we can make such a conjecture and we prefer not. Only a modeling study, dynamically interpolating between LS1 and LS2, can really proof that C1 (now in Fig.9) becomes the top right sub-mesoscale eddy, C4. We prefer not to add any speculation. We did not add any comment.

**Referee 2**
The manuscript "Marine Rapid Environmental Assessment in the Gulf of Taranto: a
multiscale approach" by Pinardi et al., is about a multiscale sampling experiment carried
out in the Gulf of Taranto to collect synoptic oceanographic data over a 10-days
period and subsequently study the thermohaline structure and the geostrophic circulation of
the area and its variability. The data analysis from the four surveys carried out
in the area from 1 to 10 October 2014 provides evidence of the large scale circulation
structure and associated mesoscale variability of the area consisting of an anticyclonic
large scale Gyre that occupies the central open ocean area of the Gulf of Taranto and
a rim current on the periphery of the gyre that undergoes large changes over the 10-
days period. Overall, I found the manuscript and the work very interesting and I think
that it deserves publication to the NHESS journal after some minor revisions are made
according to the following comments:

**Authors**
We thank the reviewer for the appreciation

**Referee**
1. Introduction section: "A new multi-scale sampling strategy was used …. the
coastal-harbor scales of Mar Grande (Fig.1). The authors should explain better the novelty of
the approach adopted to measure the T/S structure of the Gulf of Taranto.
**Authors**
We added a sentence in the Introduction, line 28 as suggested, that reads this way:
The novelty of this data collection experiment is related to the different resolution of the
stations carried out in the different areas under the strict constraint of synoptic time coverage
(3-4 days in the ocean).

**Referee**
2. Circulation structure and data collection methodology: "From a large scale point of
view : : : in Fig. 2". As the Ionian basin circulation has undergone significant interannual
changes over the period 1987 – 2013 the authors should present more evidence that the long
term average of this period is representative of the hydrodynamic situation in the Gulf of
Taranto in June and October.

**Authors**
We have modified Fig. 2 introducing the June and October 2014 reanalysis field that became
available after the cruise and that does not contain the MREA data set. It is evident that June
and August have different circulation patterns. The interannual variability of the Gulf of
Taranto circulation is outside the scope of this paper that provides the first confirmation of
reanalysis model results for October 2014. The new Fig. 2 is reproduced below and the
sentence at page 3, line 12-18 has been modified accordingly and it is reported below:
From a large scale point of view, the mean circulation in the area can be assessed by taking
the current fields from a reanalysis product (Pinardi et al., 2015) that does not contain the
MREA data. The surface circulation (Fig. 2) is anticyclonic in October 2014, while in June
2014 it is cyclonic. This opposite circulation pattern is probably connected to the different
Western Adriatic Coastal Current (WACC, Guarnieri et al., 2013), Northern Ionian Sea
outflow/inflow system in the two months and the local atmospheric forcing.. One of the major
aims of the MREA experiment was to verify the October circulation shown in Fig. 2.

[Figure]

**Fig. 2** Monthly mean surface currents from reanalysis (Pinardi et al., 2015) in the Gulf of Taranto. Top panel: June 2014. Bottom panel: October 2014. The units are m s$^{-1}$ and the color indicates the amplitude

**Referee**
3. T/S diagrams could be useful to depict the water mass structure of the area.
**Authors**
We have added a T-S diagram as new Figure 5 and a comment after line 26, page 4 which we reproduce below.
Fig. 5 shows a T-S diagram of the LS1, LS2 profiles to better identify the water masses and types. Some of the profiles extended to 900 m depth, in the central Gulf of Taranto trench (Fig.1) so that four water masses can be detected, one more with respect the three already discussed for the first 300 m. The first water mass is the surface water mass, indicated by water type 1 in Fig. 5, corresponding to low salinity and almost constant temperature. The second is the thermocline water type (number 2 in Fig. 5), due to the mixing of the surface waters and MLIW as shown clearly by the clustering of the T-S points around a line joining the two water types. Furthermore, MLIW (point 3 in Fig. 5) is now clearly detectable with a salinity and temperature increase with respect to the thermocline water mass type. Finally a deep water mass type (4 in figure 5) is also evident, with temperatures lower than 14 C and

relatively low salinities, probably of Adriatic origin.

[Figure]

**Fig. 5** T-S diagram for all the LS1-LS2 profiles, covering the depths of 1-900 m (the deepest point is in the central trench of the Gulf of Taranto, see Fig. 1). Numbers refer to the 4 water mass types found in the profiles and discussed in the text.

**Referee**
4. A more detailed discussion on the instability of the rim current is expected.
**Authors**
We have added a whole new sentence in the discussion and conclusions section since we believe this is a matter of work for the future years. The text is reproduced here:

The instability of gyre rim currents and/or large mesoscale eddy field borders has been studied in the past (Mc Williams et al., 1983, Pinardi et al., 1987, Staneva et al., 2001) and more recently for submesoscale generating fronts (Hamlington et al., 2014). The instabilities of rim currents connected to temperature frontal structures generate eddies, which are due to cyclogenenetic processes such as mixed baroclinic/barotropic instabilities. In our case the observations show that instabilities occur in a week long time and most importantly modulate the upwelling phenomena at the open ocean-shelf areas interface, a mechanism that could be very important to support good environmental conditions in the near coastal regions. Numerical modelling studies have now started to understand the vorticity and energy dynamics of the flow field observed in this experiment.

**Referee**
5. "Furthermore a precipitation event occurred between LS1 and LS2 which lowered the surface salinity of 0.1 PSU concomitantly changing the mixed layer temperatures of 0.5 C"
The authors should explain better how the precipitation event changed the mixed layer

temperatures by 0.5 C

**Authors**

Following the suggestion also of referee 1 we added a new Fig. 7 showing that together with the large precipitation event there was also an intensification of the wind which then cooled down the surface of the measured amount. We have added now a sentence at page 5, new lines 8 through 10.